# Robust Guided Diffusion for Offline Black-Box Optimization

**Can (Sam) Chen[1,2]\***, **Christopher Beckham[2,3]**, **Zixuan Liu[5]**, **Xue Liu[1,2]**, **Christopher Pal[2,3,4]**

**[1]McGill University, [2]MILA - Quebec AI Institute,**
**[3]Polytechnique Montreal, [4]Canada CIFAR AI Chair, [5]University of Washington**

Reviewed on OpenReview: `https://openreview.net/forum?id=4JcqmEZ5zt`

## Abstract

Offline black-box optimization aims to maximize a black-box function using an offline dataset of designs and their measured properties. Two main approaches have emerged: the forward approach, which learns a mapping from input to its value, thereby acting as a proxy to guide optimization, and the inverse approach, which learns a mapping from value to input for conditional generation. (a) Although proxy-free (classifier-free) diffusion shows promise in robustly modeling the inverse mapping, it lacks explicit guidance from proxies, essential for generating high-performance samples beyond the training distribution. Therefore, we propose *proxy-enhanced sampling* which utilizes the explicit guidance from a trained proxy to bolster proxy-free diffusion with enhanced sampling control. (b) Yet, the trained proxy is susceptible to out-of-distribution issues. To address this, we devise the module *diffusion-based proxy refinement*, which seamlessly integrates insights from proxy-free diffusion back into the proxy for refinement. To sum up, we propose **R**obust **G**uided **D**iffusion for Offline Black-box Optimization (**RGD**), combining the advantages of proxy (explicit guidance) and proxy-free diffusion (robustness) for effective conditional generation. RGD achieves state-of-the-art results on various design-bench tasks, underscoring its efficacy. Our code is here.

## 1 Introduction

Creating new objects to optimize specific properties is a ubiquitous challenge that spans a multitude of fields, including material science, robotic design, and genetic engineering. Traditional methods generally require interaction with a black-box function to generate new designs, a process that could be financially burdensome and potentially perilous (Hamidieh, 2018; Sarkisyan et al., 2016). Addressing this, recent research endeavors have pivoted toward a more relevant and practical context, termed offline black-box optimization (BBO) (Trabucco et al., 2022; Krishnamoorthy et al., 2023). In this context, the goal is to maximize a black-box function exclusively utilizing an offline dataset of designs and their measured properties.

There are two main approaches for this task: the forward approach and the reverse approach. The forward approach entails training a deep neural network (DNN), parameterized as $\mathcal{J}_\phi(\cdot)$, using the offline dataset. Once trained, the DNN acts as a proxy and provides explicit gradient guidance to enhance existing designs. However, this technique is susceptible to the out-of-distribution (OOD) issue, leading to potential overestimation of unseen designs and resulting in adversarial solutions (Trabucco et al., 2021).

The reverse approach aims to learn a mapping from property value to input. Inputting a high value into this mapping directly yields a high-performance design. For example, MINs (Kumar & Levine, 2020) adopts GAN (Goodfellow et al., 2014) to model this inverse mapping, and demonstrate some success. Recent works (Krishnamoorthy et al., 2023) have applied proxy-free diffusion[1] (Ho & Salimans, 2022), parameterized by $\boldsymbol{\theta}$, to model this mapping, which proves its efficacy over other generative models. Proxy-free diffusion

---

*chencan421@gmail.com/can.chen@mila.quebec.

[1]*Classifier-free diffusion* is for classification and adapted to *proxy-free diffusion* to generalize to regression.

employs a score predictor $\tilde{\boldsymbol{s}}_\theta(\cdot, \cdot, \omega)$. This represents a linear combination of conditional and unconditional scores, modulated by a strength parameter $\omega$ to balance condition and diversity in the sampling process. This guidance significantly diverges from proxy (classifier) diffusion that interprets scores as classifier gradients and thus generates adversarial solutions. Such a distinction grants proxy-free diffusion its inherent robustness in generating samples.

Nevertheless, the inverse approach, proxy-free diffusion, initially designed for in-distribution generation, such as synthesizing specific image categories, faces limitations in offline BBO. Particularly, it struggles to generate high-performance samples that exceed the training distribution due to the lack of explicit guidance[2]. Consider, for example, the optimization of a two-dimensional variable $(x_{d1}, x_{d2})$ to maximize the negative Rosenbrock function (Rosenbrock, 1960): $y(x_{d1}, x_{d2}) = -(1 - x_{d1})^2 - 100(x_{d2} - x_{d1}^2)^2$, as depicted in Figure 1. The objective is to steer the initial points (indicated in pink) towards the high-performance region (highlighted in yellow). While proxy-free diffusion can nudge the initial points closer to this high-performance region, the generated points (depicted in blue) fail to reach the high-performance region due to its lack of explicit proxy guidance.

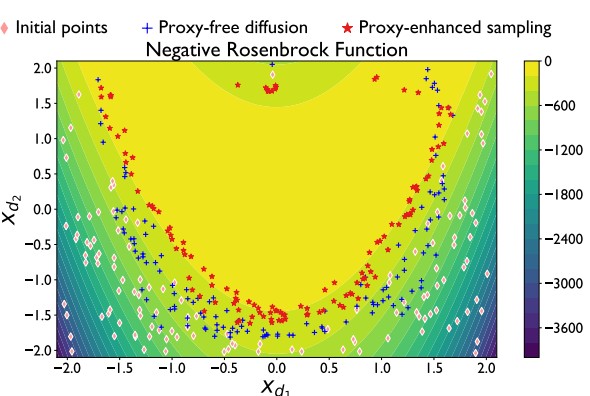

Figure 1: Motivation of explicit proxy guidance.

To address this challenge, we introduce a *proxy-enhanced sampling* module as illustrated in Figure 2(a). It incorporates the explicit guidance from the proxy $\mathcal{J}_\phi(\boldsymbol{x})$ into proxy-free diffusion to enable enhanced control over the sampling process. This module hinges on the strategic optimization of the strength parameter $\omega$ to achieve a better balance between condition and diversity, per reverse diffusion step. This incorporation not only preserves the inherent robustness of proxy-free diffusion but also leverages the explicit proxy guidance, thereby enhancing the overall conditional generation efficacy. As illustrated in Figure 1, samples (depicted in red) generated via *proxy-enhanced sampling* are more effectively guided towards, and often reach, the high-performance area (in yellow).

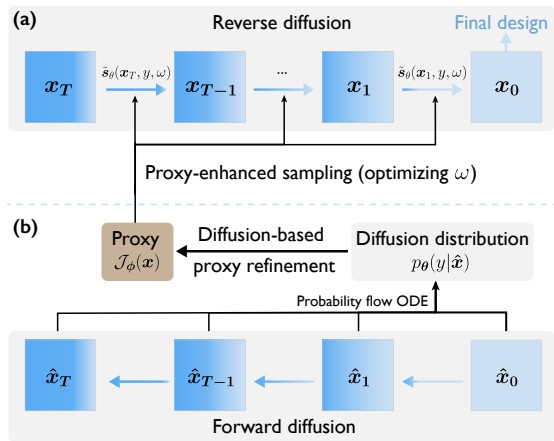

Figure 2: Overview of RGD: Module (a) incorporates proxy guidance into proxy-free diffusion to enable enhanced sampling control; Module (b) integrates insights from proxy-free diffusion back into the proxy for refinement.

Yet, the trained proxy is susceptible to out-of-distribution (OOD) issues. To address this, we devise a module *diffusion-based proxy refinement* as detailed in Figure 2(b). This module seamlessly integrates insights from proxy-free diffusion into the proxy $\mathcal{J}_\phi(\boldsymbol{x})$ for refinement. Specifically, we generate a diffusion distribution $p_{\boldsymbol{\theta}}(y|\hat{\boldsymbol{x}})$ on adversarial samples $\hat{\boldsymbol{x}}$, using the associated probability flow ODE [3]. This distribution is derived independently of a proxy, thereby exhibiting greater robustness than the proxy distribution on adversarial samples. Subsequently, we calculate the Kullback-Leibler divergence between the two distributions on adversarial samples, and use this divergence minimization as a regularization strategy to fortify the proxy's robustness and reliability.

To sum up, we propose **$\boldsymbol{R}$**obust **$\boldsymbol{G}$**uided **$\boldsymbol{D}$**iffusion for Offline Black-box Optimization (**RGD**), a novel framework that combines the advantages of proxy (explicit guidance) and proxy-free diffusion (robustness) for effective conditional generation. Our contributions are three-fold:

---

[2]Proxy-free diffusion cannot be interpreted as a proxy and thus does not provide explicit guidance (Ho & Salimans, 2022).
[3]Ordinary Differential Equation

- We propose a *proxy-enhanced sampling* module which incorporates proxy guidance into proxy-free diffusion to enable enhanced sampling control.

- We further develop *diffusion-based proxy refinement* which integrates insights from proxy-free diffusion back into the proxy for refinement.

- RGD delivers state-of-the-art performance on various design-bench tasks, emphasizing its efficacy.

## 2 Preliminaries

We provide the key notations used in this paper in Appendix H.

### 2.1 Offline Black-box Optimization

Offline black-box optimization (BBO) aims to maximize a black-box function with an offline dataset. Imagine a design space as $\mathcal{X} = \mathbb{R}^d$, where $d$ is the design dimension. The offline BBO (Trabucco et al., 2022) is:

$$\boldsymbol{x}^* = \arg \max_{\boldsymbol{x} \in \mathcal{X}} J(\boldsymbol{x}). \tag{1}$$

In this equation, $J(\cdot)$ is the unknown objective function, and $\boldsymbol{x} \in \mathcal{X}$ is a possible design. In this context, there is an offline dataset, $\mathcal{D}$, that consists of pairs of designs and their measured properties. Specifically, each $\boldsymbol{x}$ denotes a particular design, like the size of a robot, while $y$ indicates its related metric, such as its speed.

A forward approach *gradient ascent* fits a proxy distribution $p_{\boldsymbol{\phi}}(y|\boldsymbol{x}) = \mathcal{N}(J_{\boldsymbol{\phi}}(\boldsymbol{x}), \sigma_{\boldsymbol{\phi}}(\boldsymbol{x}))$ to the offline dataset where $\boldsymbol{\phi}$ denote the proxy parameters:

$$\begin{aligned} &\arg \min_{\boldsymbol{\phi}} \mathbb{E}_{(\boldsymbol{x},y) \in \mathcal{D}} [- \log p_{\boldsymbol{\phi}}(y|\boldsymbol{x})]. \\ &= \arg \min_{\boldsymbol{\phi}} \mathbb{E}_{(\boldsymbol{x},y) \in \mathcal{D}} \log(\sqrt{2\pi} \sigma_{\boldsymbol{\phi}}(\boldsymbol{x})) + \frac{(y - J_{\boldsymbol{\phi}}(\boldsymbol{x}))^2}{2\sigma_{\boldsymbol{\phi}}^2(\boldsymbol{x})}. \end{aligned} \tag{2}$$

For the sake of consistency with terminology used in the forthcoming subsection on guided diffusion, we will refer to $p_{\boldsymbol{\phi}}(\cdot|\cdot)$ as the proxy distribution and $J_{\boldsymbol{\phi}}(\cdot)$ as the proxy. Subsequently, this approach performs gradient ascent with $J_{\boldsymbol{\phi}}(\boldsymbol{x})$, leading to high-performance designs $\boldsymbol{x}^*$:

$$\boldsymbol{x}_{\tau+1} = \boldsymbol{x}_\tau + \eta \nabla_{\boldsymbol{x}} J_{\boldsymbol{\phi}}(\boldsymbol{x})|_{\boldsymbol{x}=\boldsymbol{x}_\tau}, \quad \text{for } \tau \in [0, \text{M} - 1], \tag{3}$$

converging to $\boldsymbol{x}_\text{M}$ after M steps. However, this method suffers from the out-of-distribution issue where the proxy predicts values that are notably higher than the actual values.

### 2.2 Diffusion Models

Diffusion models, a type of latent variable models, progressively introduce Gaussian noise to data in the forward process, while the reverse process aims to iteratively remove this noise through a learned score estimator Ho et al. (2020). In this work, we utilize continuous time diffusion models governed by a stochastic differential equation (SDE), as presented in Song et al. (2021). The forward SDE is formulated as:

$$d\boldsymbol{x} = \boldsymbol{f}(\boldsymbol{x}, t)dt + g(t)d\boldsymbol{w}. \tag{4}$$

where $\boldsymbol{f}(\cdot, t) : \mathbb{R}^d \to \mathbb{R}^d$ represents the drift coefficient, $g(\cdot) : \mathbb{R} \to \mathbb{R}$ denotes the diffusion coefficient and $\boldsymbol{w}$ is the standard Wiener process. This SDE transforms data distribution into noise distribution. The reverse is:

$$d\boldsymbol{x} = \left[ \boldsymbol{f}(\boldsymbol{x}, t) - g(t)^2 \nabla_{\boldsymbol{x}} \log p(\boldsymbol{x}) \right] dt + g(t)d\bar{\boldsymbol{w}}, \tag{5}$$

with $\nabla_{\boldsymbol{x}} \log p(\boldsymbol{x})$ representing the score of the marginal distribution at time $t$, and $\bar{\boldsymbol{w}}$ symbolizing the reverse Wiener process. The score function $\nabla_{\boldsymbol{x}} \log p(\boldsymbol{x})$ is estimated using a time-dependent neural network $\boldsymbol{s}_{\boldsymbol{\theta}}(\boldsymbol{x}_t, t)$, enabling us to transform noise into samples. For simplicity, we will use $\boldsymbol{s}_{\boldsymbol{\theta}}(\boldsymbol{x}_t)$, implicitly including $t$.

### 2.3 Guided Diffusion

Unconditional diffusion models capture the natural data distribution, while guided diffusion seeks to produce samples with specific desirable attributes, falling into two categories: *proxy diffusion* (Dhariwal & Nichol, 2021) and *proxy-free diffusion* (Ho & Salimans, 2022). While these were initially termed *classifier diffusion* and *classifier-free diffusion* in classification tasks, we have renamed them to *proxy diffusion* and *proxy-free diffusion*, respectively, to generalize to our regression context. Proxy diffusion combines the model's score estimate with the gradient from the proxy distribution, providing explicit guidance in line with the forward approach. However, it can be interpreted as a gradient-based adversarial attack.

In proxy-free diffusion, guidance is not dependent on proxy gradients, which enables an inherent robustness of the sampling process. Particularly, it models the score as a linear combination of an unconditional and a conditional score. A unified neural network $s_\theta(x_t, y)$ parameterizes both score types. The score $s_\theta(x_t, y)$ approximates the gradient of the log probability $\nabla_{x_t} \log p(x_t|y)$, i.e., the conditional score, while $s_\theta(x_t)$ estimates the gradient of the log probability $\nabla_{x_t} \log p(x_t)$, i.e., the unconditional score. The score function follows:

$$\tilde{s}_\theta(x_t, y, \omega) = (1 + \omega)s_\theta(x_t, y) - \omega s_\theta(x_t). \tag{6}$$

Within this context, the strength parameter $\omega$ specifies the generation's adherence to the condition $y$, which is set to the maximum value $y_{max}$ in the offline dataset following Krishnamoorthy et al. (2023). Optimization of $\omega$ balances the condition and diversity. Lower $\omega$ values increase sample diversity at the expense of conformity to $y$, and higher values do the opposite.

## 3 Related Work

**Offline black-box optimization.** A recent surge in research has presented two predominant approaches for offline BBO. The forward approach deploys a DNN to fit the offline dataset, subsequently utilizing gradient ascent to enhance existing designs. Typically, these techniques, including COMs (Trabucco et al., 2021), ROMA (Yu et al., 2021), NEMO (Fu & Levine, 2021), BDI (Chen et al., 2022b; 2023b), IOM (Qi et al., 2022a) and Parallel-mentoring (Chen et al., 2023a), are designed to embed prior knowledge within the surrogate model to alleviate the OOD issue. The reverse approach (Kumar & Levine, 2020; Chan et al., 2021) is dedicated to learning a mapping from property values back to inputs. Feeding a high value into this inverse mapping directly produces a design of elevated performance. Additionally, methods in Brookes et al. (2019); Fannjiang & Listgarten (2020) progressively tailor a generative model towards the optimized design via a proxy function and BONET (Mashkaria et al., 2023) introduces an autoregressive model trained on fixed-length trajectories to sample high-scoring designs. Recent investigations (Krishnamoorthy et al., 2023) have underscored the superiority of diffusion models in delineating the inverse mapping. However, research on specialized guided diffusion for offline BBO remains limited. This paper addresses this research gap.

**Guided diffusion.** Guided diffusion seeks to produce samples with specific desirable attributes. Contemporary research in guided diffusion primarily concentrates on enhancing the efficiency of its sampling process. Meng et al. (2023) propose a method for distilling a classifier-free guided diffusion model into a more efficient single model that necessitates fewer steps in sampling. Sadat et al. (2023) introduce the Dynamic CFG, which initially compels the model to depend more on the unconditional score and progressively shifts towards the standard CFG. However, unlike our method, it does not optimize the strength parameter. Kynkäänniemi et al. (2024) restrict the guidance to a specific range of noise levels, which improves both sampling speed and quality. Wizadwongsa & Suwajanakorn (2023) introduce an operator splitting method to expedite classifier guidance by separating the update process into two key functions: the diffusion function and the conditioning function. Additionally, Bansal et al. (2023) presents an efficient and universal guidance mechanism that utilizes a readily available proxy to enable diffusion guidance across time steps. In this work, we explore the application of guided diffusion in offline BBO, with the goal of creating tailored algorithms to efficiently generate high-performance designs.

## 4    Method

In this section, we present our method RGD, melding the strengths of proxy and proxy-free diffusion for effective conditional generation. Firstly, we describe a newly developed module termed *proxy-enhanced sampling*. It integrates explicit proxy guidance into proxy-free diffusion to enable enhanced sampling control, as detailed in Section 4.1. Subsequently, we explore *diffusion-based proxy refinement* which incorporates insights gleaned from proxy-free diffusion back into the proxy, further elaborated in Section 4.2. The overall algorithm is shown in Algorithm 1.

### 4.1    Proxy-enhanced Sampling

As discussed in Section 2.3, proxy-free diffusion trains an unconditional model and conditional models. Although proxy-free diffusion can generate samples aligned with most conditions, it traditionally lacks control due to the absence of an explicit proxy. This is particularly significant in offline BBO where we aim to obtain samples beyond the training distribution, as the offline dataset is inherently limited. Therefore, we require explicit proxy guidance to achieve enhanced sampling control. This module is outlined in Algorithm 1, Line 8- Line 16.

**Optimization of $\omega$.** Directly updating the design $\boldsymbol{x}_t$ with proxy gradient suffers from the OOD issue and determining a proper condition $y$ necessitates the manual adjustment of multiple hyperparameters (Kumar & Levine, 2020). Thus, we propose to introduce proxy guidance by only optimizing the strength parameter $\omega$ within $\tilde{\boldsymbol{s}}_{\boldsymbol{\theta}}(\boldsymbol{x}_t, y, \omega)$ in Eq. (6). As discussed in Section 2.3, the parameter $\omega$ balances the condition and diversity, and an optimized $\omega$ could achieve a better balance in the sampling process, leading to more effective generation.

---

**Algorithm 1** Robust Guided Diffusion for Offline BBO

**Input:** offline dataset $\mathcal{D}$, # of diffusion steps $T$.

1: Train proxy distribution $p_{\boldsymbol{\phi}}(y|\boldsymbol{x})$ on $\mathcal{D}$ by Eq. (2).
2: Train proxy-free diffusion model $\boldsymbol{s}_{\boldsymbol{\theta}}(\boldsymbol{x}_t, y)$ on $\mathcal{D}$.
3: */*Diffusion-based proxy refinement*/*
4: Identify adversarial samples $\hat{\boldsymbol{x}}$ via Eq.(3).
5: Compute diffusion distribution $p_{\boldsymbol{\theta}}(y|\hat{\boldsymbol{x}})$ by Eq. (12).
6: Compute KL divergence loss as per Eq. (13).
7: Refine proxy distribution $p_{\boldsymbol{\phi}}(y|\boldsymbol{x})$ through Eq. (15).
8: */*Proxy-enhanced sampling*/*
9: Begin with $\boldsymbol{x}_T \sim \mathcal{N}(\boldsymbol{0}, \boldsymbol{I})$
10: **for** $t = T - 1$ **to** $0$ **do**
11:     Derive the score $\tilde{\boldsymbol{s}}_{\boldsymbol{\theta}}(\boldsymbol{x}_{t+1}, y, \omega)$ from Eq. (6).
12:     Update $\boldsymbol{x}_{t+1}$ to $\boldsymbol{x}_t(\omega)$ using $\omega$ as per Eq. (7).
13:     Optimize $\omega$ to $\hat{\omega}$ following Eq. (8).
14:     Finalize the update of $\boldsymbol{x}_t$ with $\hat{\omega}$ via Eq. (9).
15: **end for**
16: Return $\boldsymbol{x}^* = \boldsymbol{x}_0$

---

**Enhanced Sampling.** With the score function, the update of a noisy sample $\boldsymbol{x}_{t+1}$ is computed as:

$$\boldsymbol{x}_t(\omega) = solver(\boldsymbol{x}_{t+1}, \tilde{\boldsymbol{s}}_{\boldsymbol{\theta}}(\boldsymbol{x}_{t+1}, y, \omega)), \tag{7}$$

where the *solver* is the second-order Heun solver (Süli & Mayers, 2003), chosen for its enhanced accuracy through a predictor-corrector method. A proxy is then trained to predict the property of noise $\boldsymbol{x}_t$ at time step $t$, denoted as $J_{\boldsymbol{\phi}}(\boldsymbol{x}_t, t)$. By maximizing $J_{\boldsymbol{\phi}}(\boldsymbol{x}_t(\omega), t)$ with respect to $\omega$, we can incorporate the explicit proxy guidance into proxy-free diffusion to facilitate improved sampling control in the balance between condition and diversity. This maximization process is:

$$\hat{\omega} = \omega + \eta \frac{\partial J_{\boldsymbol{\phi}}(\boldsymbol{x}_t(\omega), t)}{\partial \omega}. \tag{8}$$

where $\eta$ denotes the learning rate. We leverage the automatic differentiation capabilities of PyTorch (Paszke et al., 2019) to efficiently compute the above derivatives within the context of the solver's operation. The optimized $\hat{\omega}$ then updates the noisy sample $\boldsymbol{x}_{t+1}$ through:

$$\boldsymbol{x}_t = solver(\boldsymbol{x}_{t+1}, \tilde{\boldsymbol{s}}_{\boldsymbol{\theta}}(\boldsymbol{x}_{t+1}, y, \hat{\omega})). \tag{9}$$

This process iteratively denoises $\boldsymbol{x}_t$, utilizing it in successive steps to progressively approach $\boldsymbol{x}_0$, which represents the final high-scoring design $\boldsymbol{x}^*$.

**Proxy Training.** Notably, $J_\phi(\boldsymbol{x}_t, t)$ can be directly derived from the proxy $J_\phi(\boldsymbol{x})$, the mean of the proxy distribution $p_\phi(\cdot|\boldsymbol{x})$ in Eq. (2). This distribution is trained exclusively at the initial time step $t = 0$, eliminating the need for training across time steps and reducing computational cost. To achieve this derivation, we reverse the diffusion from $\boldsymbol{x}_t$ back to $\boldsymbol{x}_0$ using the Tweedie formula Robbins (1992):

$$\boldsymbol{x}_0 \approx \frac{\boldsymbol{x}_t + s_{\boldsymbol{\theta}}(\boldsymbol{x}_t) \cdot \sigma(t)^2}{\mu(t)}, \tag{10}$$

where $s_{\boldsymbol{\theta}}(\boldsymbol{x}_t)$ is the estimated unconditional score at time step $t$, and $\sigma(t)^2$ and $\mu(t)$ are the variance and the mean coefficient of the perturbation kernel at time $t$, as detailed in equations (32-33) in Song et al. (2021). For a more detailed derivation, refer to the Appendix B. Consequently, we express

$$J_\phi(\boldsymbol{x}_t, t) = J_\phi \left( \frac{\boldsymbol{x}_t + s_{\boldsymbol{\theta}}(\boldsymbol{x}_t) \cdot \sigma(t)^2}{\mu(t)} \right). \tag{11}$$

This formulation allows for the optimization of the strength parameter $\omega$ via Eq. (8). For simplicity, we will refer to $J_\phi(\cdot)$ in subsequent discussions.

## 4.2 Diffusion-based Proxy Refinement

In the *proxy-enhanced sampling* module, the proxy $J_\phi(\cdot)$ is employed to update the parameter $\omega$ to enable enhanced control. However, $J_\phi(\cdot)$ may still be prone to the OOD issue, especially on adversarial samples (Trabucco et al., 2021). To address this, we refine the proxy by using insights from proxy-free diffusion. The procedure of this module is specified in Algorithm 1, Lines 3-7.

**Diffusion Distribution**. Adversarial samples are identified by gradient ascent on the proxy as per Eq. (3) to form the distribution $q(\boldsymbol{x})$. We utilize a vanilla proxy to perform 300 gradient ascent steps, identifying samples with unusually high prediction scores as adversarial. This method is based on the limited extrapolation capability of the vanilla proxy, as demonstrated in Figure 3 in COMs (Trabucco et al., 2021). Consequently, these samples are vulnerable to the proxy distribution. Conversely, the proxy-free diffusion, which functions without depending on a proxy, inherently offers greater resilience against these samples, thus producing a more robust distribution. For an adversarial sample $\hat{\boldsymbol{x}} \sim q(\boldsymbol{x})$, we compute $p_{\boldsymbol{\theta}}(\hat{\boldsymbol{x}})$ and $p_{\boldsymbol{\theta}}(\hat{\boldsymbol{x}}|y)$ using the probability flow ODE associated with the SDE, as detailed in Appendix D of Song et al. (2021). The implementation can be accessed here. We estimate $p(y)$ using Gaussian kernel-density estimation. The diffusion distribution regarding $y$ is:

$$p_{\boldsymbol{\theta}}(y|\hat{\boldsymbol{x}}) = \frac{p_{\boldsymbol{\theta}}(\hat{\boldsymbol{x}}|y) \cdot p(y)}{p_{\boldsymbol{\theta}}(\hat{\boldsymbol{x}})}, \tag{12}$$

which demonstrates inherent robustness over the proxy distribution $p_\phi(y|\hat{\boldsymbol{x}})$. Yet, directly applying diffusion distribution to design optimization by gradient ascent is computationally intensive and potentially unstable due to the demands of reversing ODEs and scoring steps.

**Proxy Refinement.** We opt for a more feasible approach: refine the proxy distribution $p_\phi(y|\hat{\boldsymbol{x}}) = \mathcal{N}(J_\phi(\hat{\boldsymbol{x}}), \sigma_\phi(\hat{\boldsymbol{x}}))$ by minimizing its distance to the diffusion distribution $p_{\boldsymbol{\theta}}(y|\hat{\boldsymbol{x}})$. The distance is quantified by the Kullback-Leibler (KL) divergence:

$$\mathbb{E}_q[\text{KL}(p_\phi||p_{\boldsymbol{\theta}})] = \mathbb{E}_{q(\boldsymbol{x})} \int p_\phi(y|\hat{\boldsymbol{x}}) \log \left( \frac{p_\phi(y|\hat{\boldsymbol{x}})}{p_{\boldsymbol{\theta}}(y|\hat{\boldsymbol{x}})} \right) dy. \tag{13}$$

We avoid the reparameterization trick for minimizing this divergence as it necessitates backpropagation through $p_{\boldsymbol{\theta}}(y|\hat{\boldsymbol{x}})$, which is prohibitively expensive. Instead, for the sample $\hat{\boldsymbol{x}}$, the gradient of the KL divergence $\text{KL}(p_\phi||p_{\boldsymbol{\theta}})$ with respect to the proxy parameters $\phi$ is computed as:

$$\mathbb{E}_{p_\phi(y|\hat{\boldsymbol{x}})} \left[ \frac{d \log p_\phi(y|\hat{\boldsymbol{x}})}{d\phi} \left( 1 + \log \frac{p_\phi(y|\hat{\boldsymbol{x}})}{p_{\boldsymbol{\theta}}(y|\hat{\boldsymbol{x}})} \right) \right]. \tag{14}$$

Complete derivations are in Appendix A. The KL divergence then acts as regularization in our loss $\mathcal{L}$:

$$\mathcal{L}(\phi, \alpha) = \mathbb{E}_{\mathcal{D}}[-\log p_\phi(y|\boldsymbol{x})] + \alpha \mathbb{E}_{q(\boldsymbol{x})}[\text{KL}(p_\phi||p_{\boldsymbol{\theta}})], \tag{15}$$

where $\mathcal{D}$ is the training dataset and $\alpha$ is a hyperparameter. We propose to optimize $\alpha$ based on the validation loss via bi-level optimization as detailed in Appendix C. The computational effort of this module can exceed that of the sampling itself, a topic we explore further in Appendix E. Notably, even without this module, our method maintains strong performance, as detailed in Table 3.

## 5 Experiments

In this section, we conduct comprehensive experiments to evaluate our method's performance.

### 5.1 Benchmarks

**Tasks.** Our experiments encompass a variety of tasks, split into continuous and discrete categories.

The continuous category includes four tasks: **(1)** Superconductor (SuperC): The objective here is to engineer a superconductor composed of 86 continuous elements. The goal is to enhance the critical temperature using $17,010$ design samples. This task is based on the dataset from Hamidieh (2018). **(2)** Ant Morphology (Ant): In this task, the focus is on developing a quadrupedal ant robot, comprising 60 continuous parts, to augment its crawling velocity. It uses $10,004$ design instances from the dataset in Trabucco et al. (2022); Brockman et al. (2016). **(3)** D'Kitty Morphology (D'Kitty): Similar to Ant Morphology, this task involves the design of a quadrupedal D'Kitty robot with 56 components, aiming to improve its crawling speed with $10,004$ designs, as described in Trabucco et al. (2022); Ahn et al. (2020). **(4)** Rosenbrock (Rosen): The aim of this task is to optimize a 60-dimension continuous vector to maximize the Rosenbrock black-box function. It uses 50000 designs from the low-scoring part (Rosenbrock, 1960).

For the discrete category, we explore three tasks: **(1)** TF Bind 8 (TF8): The goal is to identify an 8-unit DNA sequence that maximizes binding activity. This task uses $32,898$ designs and is detailed in Barrera et al. (2016). **(2)** TF Bind 10 (TF10): Similar to TF8, but with a 10-unit DNA sequence and a larger pool of $50,000$ samples, as described in (Barrera et al., 2016). **(3)** Neural Architecture Search (NAS): This task focuses on discovering the optimal neural network architecture to improve test accuracy on the CIFAR-10 dataset, using $1,771$ designs (Zoph & Le, 2017).

**Evaluation.** In this study, we utilize the oracle evaluation from design-bench (Trabucco et al., 2022). Adhering to this established protocol, we analyze the top 128 promising designs from each method. The evaluation metric employed is the $100^{th}$ percentile normalized ground-truth score, calculated using the formula $y_n = \frac{y - y_{\min}}{y_{\max} - y_{\min}}$, where $y_{\min}$ and $y_{\max}$ signify the lowest and highest scores respectively in the comprehensive, yet unobserved, dataset. In addition to these scores, we provide an overview of each method's effectiveness through the mean and median rankings across all evaluated tasks. Notably, the best design discovered in the offline dataset, designated as $\mathcal{D}(\textbf{best})$, is also included for reference. For further details on the $50^{th}$ percentile (median) scores, please refer to Appendix C.

### 5.2 Method Comparison

Our approach is evaluated against two primary groups of baseline methods: forward and inverse approaches. Forward approaches enhance existing designs through gradient ascent. This includes: **(i)** Grad: utilizes simple gradient ascent on current designs for new creations; **(ii)** ROMA (Yu et al., 2021): implements smoothness regularization on proxies; **(iii)** COMs (Trabucco et al., 2021): applies regularization to assign lower scores to adversarial designs; **(iv)** NEMO (Fu & Levine, 2021): bridges the gap between proxy and actual functions using normalized maximum likelihood; **(v)** BDI (Chen et al., 2022b): utilizes both forward and inverse mappings to transfer knowledge from offline datasets to the designs; **(vi)** IOM (Qi et al., 2022b): ensures consistency between representations of training datasets and optimized designs.

Inverse approaches focus on learning a mapping from a design's property value back to its input. High property values are input into this inverse mapping to yield enhanced designs. This includes: **(i)** CbAS (Brookes et al., 2019): CbAS employs a VAE model to implicitly implement the inverse mapping. It gradually tunes its distribution toward higher scores by raising the scoring threshold. This process can be interpreted as incrementally increasing the conditional score within the inverse mapping framework. **(ii)** Autofocused CbAS

(Auto.CbAS) (Fannjiang & Listgarten, 2020): adopts importance sampling for retraining a regression model based on CbAS. **(iii)** MIN (Kumar & Levine, 2020): maps scores to designs via a GAN model and explore this mapping for optimal designs. **(iv)** BONET (Mashkaria et al., 2023): introduces an autoregressive model for sampling high-scoring designs. **(v)** DDOM (Krishnamoorthy et al., 2023): utilizes proxy-free diffusion to model the inverse mapping.

Traditional methods as detailed in Trabucco et al. (2022) are also considered: **(i)** CMA-ES (Hansen, 2006): modifies the covariance matrix to progressively shift the distribution towards optimal designs; **(ii)** BO-qEI (Wilson et al., 2017): implements Bayesian optimization to maximize the proxy and utilizes the quasi-Expected-Improvement acquisition function for design suggestion, labeling designs using the proxy; **(iii)** REINFORCE (Williams, 1992): enhances the input space distribution using the learned proxy model.

## 5.3 Experimental Configuration

In alignment with the experimental protocols established in Trabucco et al. (2022); Chen et al. (2022b), we have tailored our training methodologies for all approaches, utilizing a three-layer MLP architecture for all involved proxies. For methods such as BO-qEI, CMA-ES, REINFORCE, CbAS, and Auto.CbAS that do not utilize gradient ascent, we base our approach on the findings reported in Trabucco et al. (2022). We adopted $T = 1000$ diffusion sampling steps, set the condition $y$ to $y_{max}$, and initial strength $\omega$ as 2 in line with Krishnamoorthy et al. (2023). To ensure reliability and consistency in our comparative analysis, each experimental setting was replicated across 8 independent runs, unless stated otherwise, with the presentation of both mean values and standard deviations. These experiments were conducted using a NVIDIA GeForce V100 GPU. We have detailed the computational overhead of our approach in Appendix E to provide a comprehensive view of its practicality.

Table 1: Results (maximum normalized score, $mean \pm std$) on continuous tasks.

| Method | Superconductor | Ant Morphology | D'Kitty Morphology | Rosenbrock |
|---|---|---|---|---|
| $\mathcal{D}(\textbf{best})$ | 0.399 | 0.565 | 0.884 | 0.518 |
| BO-qEI | $0.402 \pm 0.034$ | $0.819 \pm 0.000$ | $0.896 \pm 0.000$ | $0.772 \pm 0.012$ |
| CMA-ES | $0.465 \pm 0.024$ | $\textbf{1.214} \pm \textbf{0.732}$ | $0.724 \pm 0.001$ | $0.470 \pm 0.026$ |
| REINFORCE | $0.481 \pm 0.013$ | $0.266 \pm 0.032$ | $0.562 \pm 0.196$ | $0.558 \pm 0.013$ |
| Grad | $0.490 \pm 0.009$ | $0.932 \pm 0.015$ | $0.930 \pm 0.002$ | $0.701 \pm 0.092$ |
| COMs | $\textbf{0.504} \pm \textbf{0.022}$ | $0.818 \pm 0.017$ | $0.905 \pm 0.017$ | $0.672 \pm 0.075$ |
| ROMA | $\textbf{0.507} \pm \textbf{0.013}$ | $0.898 \pm 0.029$ | $0.928 \pm 0.007$ | $0.663 \pm 0.072$ |
| NEMO | $0.499 \pm 0.003$ | $0.956 \pm 0.013$ | $\textbf{0.953} \pm \textbf{0.010}$ | $0.614 \pm 0.000$ |
| IOM | $\textbf{0.524} \pm \textbf{0.022}$ | $0.929 \pm 0.037$ | $0.936 \pm 0.008$ | $0.712 \pm 0.068$ |
| BDI | $\textbf{0.513} \pm \textbf{0.000}$ | $0.906 \pm 0.000$ | $0.919 \pm 0.000$ | $0.630 \pm 0.000$ |
| CbAS | $\textbf{0.503} \pm \textbf{0.069}$ | $0.876 \pm 0.031$ | $0.892 \pm 0.008$ | $0.702 \pm 0.008$ |
| Auto.CbAS | $0.421 \pm 0.045$ | $0.882 \pm 0.045$ | $0.906 \pm 0.006$ | $0.721 \pm 0.007$ |
| MIN | $0.499 \pm 0.017$ | $0.445 \pm 0.080$ | $0.892 \pm 0.011$ | $0.702 \pm 0.074$ |
| BONET | $0.422 \pm 0.019$ | $0.925 \pm 0.010$ | $0.941 \pm 0.001$ | $0.780 \pm 0.009$ |
| DDOM | $0.495 \pm 0.012$ | $0.940 \pm 0.004$ | $0.935 \pm 0.001$ | $\textbf{0.789} \pm \textbf{0.003}$ |
| $\textbf{\textit{RGD}}$ | $\textbf{0.515} \pm \textbf{0.011}$ | $\textbf{0.968} \pm \textbf{0.006}$ | $\textbf{0.943} \pm \textbf{0.004}$ | $\textbf{0.797} \pm \textbf{0.011}$ |

## 5.4 Results and Analysis

In Tables 1 and 2, we showcase our experimental results for both continuous and discrete tasks. To clearly differentiate among the various approaches, distinct lines separate traditional, forward, and inverse approaches within the tables For every task, algorithms performing within a standard deviation of the highest score are emphasized by **bolding** following Trabucco et al. (2021).

We make the following observations. (1) As highlighted in Table 2, RGD not only achieves the top rank but also demonstrates the best performance in six out of seven tasks, emphasizing the robustness and superiority of our method. (2) RGD outperforms the VAE-based CbAS, the GAN-based MIN and the Transformer-based

Table 2: Results (maximum normalized score, *mean ± std*) on discrete tasks & ranking on all tasks.

| Method | TF Bind 8 | TF Bind 10 | NAS | Rank Mean | Rank Median |
|---|---|---|---|---|---|
| $\mathcal{D}(\mathbf{best})$ | 0.439 | 0.467 | 0.436 | | |
| BO-qEI | $0.798 \pm 0.083$ | $0.652 \pm 0.038$ | $\mathbf{1.079 \pm 0.059}$ | 9.1/15 | 11/15 |
| CMA-ES | $0.953 \pm 0.022$ | $0.670 \pm 0.023$ | $0.985 \pm 0.079$ | 7.3/15 | 4/15 |
| REINFORCE | $0.948 \pm 0.028$ | $0.663 \pm 0.034$ | $-1.895 \pm 0.000$ | 11.3/15 | 14/15 |
| Grad | $0.872 \pm 0.062$ | $0.646 \pm 0.052$ | $0.624 \pm 0.102$ | 9.0/15 | 10/15 |
| COMs | $0.517 \pm 0.115$ | $0.613 \pm 0.003$ | $0.783 \pm 0.029$ | 10.3/15 | 10/15 |
| ROMA | $0.927 \pm 0.033$ | $0.676 \pm 0.029$ | $0.927 \pm 0.071$ | 6.1/15 | 6/15 |
| NEMO | $0.942 \pm 0.003$ | $\mathbf{0.708 \pm 0.022}$ | $0.737 \pm 0.010$ | 5.3/15 | 5/15 |
| IOM | $0.823 \pm 0.130$ | $0.650 \pm 0.042$ | $0.559 \pm 0.081$ | 7.4/15 | 6/15 |
| BDI | $0.870 \pm 0.000$ | $0.605 \pm 0.000$ | $0.722 \pm 0.000$ | 9.6/15 | 9/15 |
| CbAS | $0.927 \pm 0.051$ | $0.651 \pm 0.060$ | $0.683 \pm 0.079$ | 8.7/15 | 8/15 |
| Auto.CbAS | $0.910 \pm 0.044$ | $0.630 \pm 0.045$ | $0.506 \pm 0.074$ | 10.3/15 | 10/15 |
| MIN | $0.905 \pm 0.052$ | $0.616 \pm 0.021$ | $0.717 \pm 0.046$ | 10.4/15 | 10/15 |
| BONET | $0.913 \pm 0.008$ | $0.621 \pm 0.030$ | $0.724 \pm 0.008$ | 7.7/15 | 8/15 |
| DDOM | $0.957 \pm 0.006$ | $0.657 \pm 0.006$ | $0.745 \pm 0.070$ | 4.9/15 | 5/15 |
| *RGD* | $\mathbf{0.974 \pm 0.003}$ | $\mathbf{0.694 \pm 0.018}$ | $0.825 \pm 0.063$ | $\mathbf{2.0/15}$ | $\mathbf{2/15}$ |

BONET. This result highlights the superiority of diffusion models in modeling inverse mappings compared to other generative approaches. (3) Upon examining TF Bind 8, we observe that the average rankings for forward and inverse methods stand at 10.3 and 6.0, respectively. In contrast, for TF Bind 10, both methods have the same average ranking of 8.7, indicating no advantage. This notable advantage of inverse methods in TF Bind 8 implies that the relatively smaller design space of TF Bind 8 ($4^8$) facilitates easier inverse mapping, as opposed to the more complex space in TF Bind 10 ($4^{10}$). (4) RGD's performance is less impressive on NAS, where designs are encoded as 64-length sequences of 5-category one-hot vectors. This may stem from the design-bench's encoding not fully capturing the sequential and hierarchical aspects of network architectures, affecting the efficacy of inverse mapping modeling.

## 5.5 Ablation Studies

In this section, we present a series of ablation studies to scrutinize the individual contributions of distinct components in our methodology. We employ our proposed approach as a benchmark and methodically exclude key modules, such as the *proxy-enhanced sampling* and *diffusion-based proxy refinement*, to assess their influence on performance. These variants are denoted as *w/o proxy-e* and *w/o diffusion-b r*. Additionally, we explore the strategy of directly performing gradient ascent on the diffusion intermediate state with a learning rate of 0.1, referred to as *direct grad update*. The results from these ablations are detailed in Table 3.

Our analysis reveals that omitting either module results in a decrease in performance, thereby affirming the importance of each component. The *w/o diffusion-b r* variant generally surpasses *w/o proxy-e*, highlighting the utility of the proxy-enhanced sampling even with a basic proxy setup. Conversely, *direct grad update* tends to produce subpar results across tasks, likely attributable to the proxy's limitations in handling out-of-distribution samples, leading to suboptimal design optimizations.

To further dive into the proxy-enhanced sampling module, we visualize the strength ratio $\omega/\omega_0$—where $\omega_0$ represents the initial strength—across diffusion steps $t$. This analysis is depicted in Figure 3 for two tasks: Ant and TF10. We observe a pattern of initial decrease followed by an increase in $\omega$ across both tasks. This pattern can be interpreted as follows: The decrease in $\omega$ facilitates the generation of a more diverse set of samples, enhancing exploratory capabilities. Subsequently, the increase in $\omega$ signifies a shift towards integrating high-performance features into the sample generation. Within this context, conditioning on the maximum $y$ is not aimed at achieving the dataset's maximum but at enriching samples with high-scoring attributes. Overall, this adjustment of $\omega$ effectively balances between generating novel solutions and honing in on high-quality ones.

Table 3: Ablation studies on RGD (maximum normalized score, $mean \pm std$).

| Task | D | RGD | w/o proxy-e | w/o diffusion-b r | direct grad update |
|------|---|-----|-------------|-------------------|--------------------|
| SuperC | 86 | **0.515 ± 0.011** | 0.495 ± 0.012 | 0.502 ± 0.005 | 0.456 ± 0.002 |
| Ant | 60 | **0.968 ± 0.006** | 0.940 ± 0.004 | 0.961 ± 0.011 | −0.006 ± 0.003 |
| D'Kitty | 56 | **0.943 ± 0.004** | 0.935 ± 0.001 | 0.939 ± 0.003 | 0.714 ± 0.001 |
| Rosen | 60 | 0.797 ± 0.011 | 0.789 ± 0.003 | **0.813 ± 0.005** | 0.241 ± 0.283 |
| TF8 | 8 | **0.974 ± 0.003** | 0.957 ± 0.007 | 0.960 ± 0.006 | 0.905 ± 0.000 |
| TF10 | 10 | **0.694 ± 0.018** | 0.657 ± 0.006 | 0.667 ± 0.009 | 0.672 ± 0.018 |
| NAS | 64 | **0.825 ± 0.063** | 0.745 ± 0.070 | 0.717 ± 0.032 | 0.718 ± 0.032 |

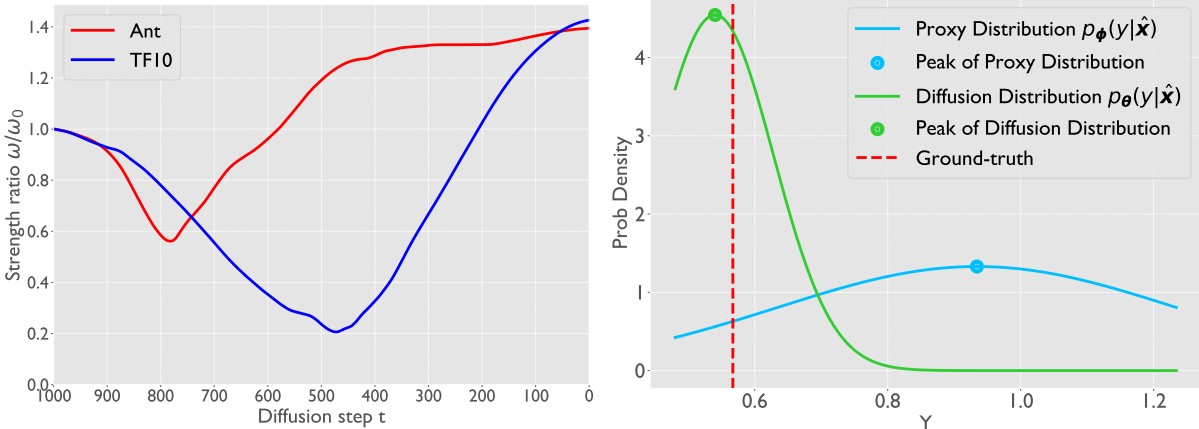

Figure 3: This adjustment of $\omega$ effectively balances between generating novel solutions and honing in on high-quality ones during sampling.

Figure 4: The proxy distribution overestimates the ground truth, while the diffusion distribution closely aligns with it, demonstrating its robustness.

In addition, we visualize the proxy distribution alongside the diffusion distribution for a sample $\hat{x}$ from the Ant task in Figure 4, to substantiate the efficacy of diffusion-based proxy refinement. The proxy distribution significantly overestimates the ground truth, whereas the diffusion distribution closely aligns with it, demonstrating the robustness of diffusion distribution. For a more quantitative analysis, we compute the expectation of both distributions and compare them with the ground truth. The mean of the diffusion distribution is calculated as $\mathbb{E}_{p_{\theta}(y|\hat{x})}[y] = \mathbb{E}_{p_{\phi}(y|\hat{x})}\left[\frac{p_{\theta}(y|\hat{x})}{p_{\phi}(y|\hat{x})}y\right]$. The MSE loss for the proxy distribution is 2.88, while for the diffusion distribution, it is 0.13 on the Ant task. Additionally, we evaluate this on the TFB10 task, where the MSE loss for the proxy distribution is 323.63 compared to 0.82 for the diffusion distribution. These results further corroborate the effectiveness of our proposed module.

Furthermore, we (1) investigate the impact of replacing our trained proxy model with alternative approaches, specifically ROMA and COMs, (2) analyze the performance with an optimized condition $y$ and (3) explore a simple annealing approach of $\omega$. For a comprehensive discussion on these, readers are referred to Appendix F, where the results further highlight the effectiveness of our trained proxy and the $\omega$ adaptation strategy.

### 5.6 Hyperparameter Sensitivity Analysis

This section investigates the sensitivity of $RGD$ to various hyperparameters. Specifically, we analyze the effects of (1) the number of diffusion sampling steps $T$, (2) the condition $y$, and (3) the learning rate $\eta$ of the proxy-enhanced sampling. These parameters are evaluated on two tasks: the continuous Ant task and the discrete TFB10 task. Our method is generally robust to these hyperparameters. For a detailed discussion, see Appendix G.

## 6 Conclusion

In conclusion, we propose **R**obust **G**uided **D**iffusion for Offline Black-box Optimization (**RGD**). The *proxy-enhanced sampling* module adeptly integrates proxy guidance to facilitate improved sampling control, while the *diffusion-based proxy refinement* module leverages proxy-free diffusion insights for proxy improvement. Empirical evaluations on design-bench have showcased RGD's outstanding performance, further validated by ablation studies on the contributions of these novel components. We discuss the broader impact and limitation in Appendix I.

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

## A    Derivation

This section provides a derivation of the gradient of the KL divergence. Let's consider the KL divergence term, defined as:

$$\text{KL}(p_{\boldsymbol{\phi}}||p_{\boldsymbol{\theta}}) = \int p_{\boldsymbol{\phi}}(y|\hat{\boldsymbol{x}}) \log \left( \frac{p_{\boldsymbol{\phi}}(y|\hat{\boldsymbol{x}})}{p_{\boldsymbol{\theta}}(y|\hat{\boldsymbol{x}})} \right) dy. \tag{16}$$

The gradient with respect to the parameters $\boldsymbol{\phi}$ is computed as follows:

$$\begin{aligned}
\frac{d\text{KL}(p_{\boldsymbol{\phi}}||p_{\boldsymbol{\theta}})}{d\boldsymbol{\phi}} &= \int \frac{dp_{\boldsymbol{\phi}}(y|\hat{\boldsymbol{x}})}{d\boldsymbol{\phi}} \left( 1 + \log \frac{p_{\boldsymbol{\phi}}(y|\hat{\boldsymbol{x}})}{p_{\boldsymbol{\theta}}(y|\hat{\boldsymbol{x}})} \right) dy \\
&= \int p_{\boldsymbol{\phi}}(y|\hat{\boldsymbol{x}}) \frac{d\log p_{\boldsymbol{\phi}}(y|\hat{\boldsymbol{x}})}{d\boldsymbol{\phi}} (1 + \log \frac{p_{\boldsymbol{\phi}}(y|\hat{\boldsymbol{x}})}{p_{\boldsymbol{\theta}}(y|\hat{\boldsymbol{x}})}) dy \\
&= \mathbb{E}_{p_{\boldsymbol{\phi}}(y|\hat{\boldsymbol{x}})} \left[ \frac{d\log p_{\boldsymbol{\phi}}(y|\hat{\boldsymbol{x}})}{d\boldsymbol{\phi}} \left( 1 + \log \frac{p_{\boldsymbol{\phi}}(y|\hat{\boldsymbol{x}})}{p_{\boldsymbol{\theta}}(y|\hat{\boldsymbol{x}})} \right) \right].
\end{aligned} \tag{17}$$

## B    Proxy Training

We follow Eq.(33) from Song et al. (2021) where $p(\mathbf{x}_t|\mathbf{x}_0) = \mathcal{N}(\mathbf{x}_t; \mu(t)\mathbf{x}_0, \sigma^2(t)I)$. Given this, we can sample $\mathbf{x}_t$ from $\mathbf{x}_0$ using: $\mathbf{x}_t = \mu(t)\mathbf{x}_0 + \epsilon\sigma(t)$. To recover $\mathbf{x}_0$ from $\mathbf{x}_t$, we need to know $\epsilon$, which approximates as $\epsilon \approx -\sigma(t) \cdot s_\theta(\mathbf{x}_t)$. Using this approximation, we derive $\mathbf{x}_0 = \frac{\mathbf{x}_t - \epsilon\sigma(t)}{\mu(t)} \approx \frac{\mathbf{x}_t + \sigma^2(t) \cdot s_\theta(\mathbf{x}_t)}{\mu(t)}$

This approach originates from Song et al. (2021), and we utilize the implementation framework detailed in another seminal work (Huang et al., 2021). Our code, available here, implements this process as follows:

- Line 24 implements $\mu(t)$

- Line 27 implements $\sigma^2(t)$

- Line 37 describes the sampling process: $\mathbf{x}_t = \mu(t)\mathbf{x}_0 + \epsilon\sigma(t)$

- Line 112 optimizes: $\epsilon \approx -\sigma(t) \cdot s_\theta(\mathbf{x}_t)$, where $\epsilon$ is the target, $\sigma(t)$ is the std, and $a$ is $s_\theta(\mathbf{x}_t)$.

Additionally, it's worth noting that Eq.( 10) in our work aligns closely with Eq.(15) from the seminal work DDPM (Ho et al., 2020). In DDPM, they present the equation $x_0 \approx \frac{x_t - \sqrt{1-\alpha_t}\epsilon_\theta(x_t)}{\sqrt{\alpha_t}}$, which is derived in a discrete setting.

## C    Hyperparameter Optimization

We propose adjusting $\alpha$ based on the validation loss, establishing a bi-level optimization framework:

$$\alpha^* = \arg\min_\alpha \mathbb{E}_{\mathcal{D}_v}[\log p_{\boldsymbol{\phi}^*(\alpha)}(y_v|\boldsymbol{x}_v)], \tag{18}$$

$$\text{s.t.} \quad \boldsymbol{\phi}^*(\alpha) = \arg\min_{\boldsymbol{\phi}} \mathcal{L}(\boldsymbol{\phi}, \alpha). \tag{19}$$

Within this context, $\mathcal{D}_v$ represents the validation dataset sampled from the offline dataset. The inner optimization task, which seeks the optimal $\boldsymbol{\phi}^*(\alpha)$, is efficiently approximated via first-order gradient descent methods. We use batch optimization, with each batch containing 256 training samples and 256 validation samples. The bi-level optimization process updates the hyperparameter with a single iteration for both the inner and outer levels.

## D    Evaluation of Median Scores

While the main text of our paper focuses on the $100^{th}$ percentile scores, this section provides an in-depth analysis of the $50^{th}$ percentile scores. These median scores, previously explored in Trabucco et al. (2022), serve as an additional metric to assess the performance of our *RGD* method. The outcomes for continuous tasks are detailed in Table 6, and those pertaining to discrete tasks, along with their respective ranking statistics, are outlined in Table 7. An examination of Table 7 highlights the notable success of the *RGD* approach, as it achieves the top rank in this evaluation. This finding underscores the method's robustness and effectiveness.

## E    Computational Overhead

Table 4: Computational Overhead (in seconds).

| Process | SuperC | Ant | D'Kitty | NAS |
|---|---|---|---|---|
| Proxy training | 40.8 | 74.5 | 24.7 | 7.8 |
| Diffusion training | 405.9 | 767.9 | 251.1 | 56.0 |
| Proxy-e sampling | 30.0 | 29.7 | 29.6 | 31.5 |
| Diffusion-b proxy r | 3104.6 | 4036.7 | 2082.8 | 3096.2 |
| Overall cost | 3581.3 | 4908.8 | 2388.2 | 3191.5 |

In this section, we analyze the computational overhead of our method. RGD consists of two core components: proxy-enhanced sampling (*proxy-e sampling*) and diffusion-based proxy refinement (*diffusion-b proxy r*). Additionally, RGD employs a trained proxy and a proxy-free diffusion model, whose computational demands are denoted as *proxy training* and *diffusion training*, respectively.

Table 4 indicates that experiments can be completed within approximately one hour, demonstrating efficiency. The *diffusion-based proxy refinement* module is the primary contributor to the computational overhead, primarily due to the usage of a probability flow ODE for sample likelihood computation. However, as this is a one-time process for refining the proxy, its high computational cost is offset by its non-recurring nature. We have also compared the time costs of various competitive methods for the 86-dimension continuous SuperC task. Our method (RGD) requires approximately 3581.3 seconds, Auto.CbAS requires 425.2 seconds, MIN requires 921.4 seconds, BONET requires 673.2 seconds, DDOM requires 460.7 seconds, and BDI requires 618.4 seconds. For the discrete 64-dimension NAS task, our method (RGD) takes approximately 3191.5 seconds, while Auto.CbAS takes 389.3 seconds, MIN takes 879.1 seconds, BONET takes 485.3 seconds, DDOM takes 103.4 seconds, and BDI takes 498.0 seconds. The evaluation of any mentioned method in NAS entails training the CIFAR-10 dataset over 20 epochs for 128 architectural designs, accumulating a total of 153.6 hours. In comparison, the one-hour computation time of our method appears negligible. This comparative analysis illustrates the computational overhead of RGD relative to other methods.

The computational bottleneck of our method is the *diffusion-based proxy refinement* module. When we remove this module, this adjustment significantly reduces computational overhead: from 3581.3 seconds to 476.7 seconds on SuperC, and from 3191.5 seconds to 95.3 seconds on NAS, rendering our method more efficient than the comparison methods. Following this adjustment, we recalculate the rankings based on the results presented in Tables 1, 2, and 3. The new ranking, as shown in the Table 5, reaffirms that our method continues to hold its position as the top-performing method in terms of ranking.

In contexts such as robotics or bio-chemical research, the most time-intensive part of the production cycle is usually the evaluation of the unknown objective function. Therefore, the time differences between methods for deriving high-performance designs are less critical in actual production environments, highlighting RGD's practicality where optimization performance are prioritized over computational speed. This aligns with recent literature (A.3 Computational Complexity in Chen et al. (2022a) and A.7.5. Computational Cost in Chen et al. (2023b)) indicating that in black-box optimization scenarios, computational time is relatively minor compared to the time and resources dedicated to experimental validation phases.

Table 5: Ranking Statistics of Methods.

| Method | Rank Mean | Rank Median |
|--------|-----------|-------------|
| BO-qEI | 9.14 | 11.00 |
| CMA-ES | 7.14 | **3.00** |
| REINFORCE | 11.29 | 14.00 |
| Grad | 9.00 | 10.00 |
| COMs | 10.00 | 10.00 |
| ROMA | 5.86 | 6.00 |
| NEMO | 5.14 | 5.00 |
| IOM | 7.43 | 6.00 |
| BDI | 9.29 | 8.00 |
| CbAS | 8.57 | 8.00 |
| Auto.CbAS | 10.29 | 10.00 |
| MIN | 10.29 | 10.00 |
| BONET | 7.43 | 7.00 |
| DDOM | 4.71 | 5.00 |
| RGD | **3.71** | **3.00** |

Table 6: Results (median normalized score, $mean \pm std$) on continuous tasks.

| Method | Superconductor | Ant Morphology | D'Kitty Morphology | Rosenbrock |
|--------|----------------|----------------|--------------------|------------|
| BO-qEI | $0.300 \pm 0.015$ | $0.567 \pm 0.000$ | $\mathbf{0.883 \pm 0.000}$ | $\mathbf{0.761 \pm 0.004}$ |
| CMA-ES | $0.379 \pm 0.003$ | $-0.045 \pm 0.004$ | $0.684 \pm 0.016$ | $0.200 \pm 0.000$ |
| REINFORCE | $\mathbf{0.463 \pm 0.016}$ | $0.138 \pm 0.032$ | $0.356 \pm 0.131$ | $0.553 \pm 0.008$ |
| Grad | $0.339 \pm 0.013$ | $0.532 \pm 0.014$ | $0.867 \pm 0.006$ | $0.540 \pm 0.025$ |
| COMs | $0.312 \pm 0.018$ | $0.568 \pm 0.002$ | $\mathbf{0.883 \pm 0.000}$ | $0.419 \pm 0.286$ |
| ROMA | $0.364 \pm 0.020$ | $0.467 \pm 0.031$ | $0.850 \pm 0.006$ | $-0.121 \pm 0.242$ |
| NEMO | $0.319 \pm 0.010$ | $0.592 \pm 0.001$ | $\mathbf{0.882 \pm 0.002}$ | $0.510 \pm 0.000$ |
| IOM | $0.343 \pm 0.018$ | $0.513 \pm 0.024$ | $0.873 \pm 0.009$ | $0.126 \pm 0.443$ |
| BDI | $0.412 \pm 0.000$ | $0.474 \pm 0.000$ | $0.855 \pm 0.000$ | $0.561 \pm 0.000$ |
| CbAS | $0.111 \pm 0.017$ | $0.384 \pm 0.016$ | $0.753 \pm 0.008$ | $0.676 \pm 0.008$ |
| Auto.CbAS | $0.131 \pm 0.010$ | $0.364 \pm 0.014$ | $0.736 \pm 0.025$ | $0.695 \pm 0.008$ |
| MIN | $0.336 \pm 0.016$ | $0.618 \pm 0.040$ | $\mathbf{0.887 \pm 0.004}$ | $0.634 \pm 0.082$ |
| BONET | $0.319 \pm 0.014$ | $0.615 \pm 0.004$ | $\mathbf{0.895 \pm 0.021}$ | $0.630 \pm 0.009$ |
| DDOM | $0.295 \pm 0.001$ | $0.590 \pm 0.003$ | $0.870 \pm 0.001$ | $0.640 \pm 0.001$ |
| ***RGD*** | $0.308 \pm 0.003$ | $\mathbf{0.684 \pm 0.006}$ | $\mathbf{0.874 \pm 0.001}$ | $0.644 \pm 0.002$ |

## F Further Ablation Studies

In this section, we extend our exploration to include alternative proxy refinement schemes, namely ROMA and COMs, to compare against our diffusion-based proxy refinement module. The objective is to assess the relative effectiveness of these schemes in the context of the Ant and TFB10 tasks. The comparative results are presented in Table 8. Our investigation reveals that proxies refined through ROMA and COMs exhibit performance akin to the vanilla proxy and they fall short of achieving the enhancements seen with our diffusion-based proxy refinement. We hypothesize that the diffusion-based proxy refinement, by aligning closely with the characteristics of the diffusion model, provides a more relevant and impactful signal. This alignment improves the proxy's ability to enhance the sampling process more effectively.

Additionally, we contrast our approach, which adjusts the strength parameter $\omega$, with the MIN method that focuses on identifying an optimal condition $y$. The MIN strategy entails optimizing a Lagrangian objective with respect to $y$, a process that requires manual tuning of four hyperparameters. We adopt their methodology to determine optimal conditions $y$ and incorporate these into the proxy-free diffusion for tasks Ant and TF10. The normalized scores for Ant and TF10 are $0.950 \pm 0.017$ and $0.660 \pm 0.027$, respectively. The outcomes

Table 7: Results (median normalized score, $mean \pm std$) on discrete tasks & ranking on all tasks.

| Method | TF Bind 8 | TF Bind 10 | NAS | Rank Mean | Rank Median |
|---|---|---|---|---|---|
| BO-qEI | $0.439 \pm 0.000$ | $0.467 \pm 0.000$ | $\mathbf{0.544 \pm 0.099}$ | 6.4/15 | 7/15 |
| CMA-ES | $0.537 \pm 0.014$ | $0.484 \pm 0.014$ | $\mathbf{0.591 \pm 0.102}$ | 8.0/15 | 5/15 |
| REINFORCE | $0.462 \pm 0.021$ | $0.475 \pm 0.008$ | $-1.895 \pm 0.000$ | 9.7/15 | 9/15 |
| Grad | $0.546 \pm 0.022$ | $0.526 \pm 0.029$ | $0.443 \pm 0.126$ | 6.6/15 | 8/15 |
| COMs | $0.439 \pm 0.000$ | $0.467 \pm 0.000$ | $\mathbf{0.529 \pm 0.003}$ | 7.7/15 | 8/15 |
| ROMA | $0.543 \pm 0.017$ | $0.518 \pm 0.024$ | $\mathbf{0.529 \pm 0.008}$ | 7.6/15 | 5/15 |
| NEMO | $0.436 \pm 0.016$ | $0.453 \pm 0.013$ | $\mathbf{0.563 \pm 0.020}$ | 8.3/15 | 8/15 |
| IOM | $0.439 \pm 0.000$ | $0.474 \pm 0.014$ | $-0.083 \pm 0.012$ | 9.3/15 | 8/15 |
| BDI | $0.439 \pm 0.000$ | $0.476 \pm 0.000$ | $\mathbf{0.517 \pm 0.000}$ | 7.3/15 | 8/15 |
| CbAS | $0.428 \pm 0.010$ | $0.463 \pm 0.007$ | $0.292 \pm 0.027$ | 11.3/15 | 12/15 |
| Auto.CbAS | $0.419 \pm 0.007$ | $0.461 \pm 0.007$ | $0.217 \pm 0.005$ | 11.9/15 | 13/15 |
| MIN | $0.421 \pm 0.015$ | $0.468 \pm 0.006$ | $0.433 \pm 0.000$ | 7.0/15 | 7/15 |
| BONET | $0.507 \pm 0.007$ | $0.460 \pm 0.013$ | $\mathbf{0.571 \pm 0.095}$ | 5.9/15 | 6/15 |
| DDOM | $0.553 \pm 0.002$ | $0.488 \pm 0.001$ | $0.367 \pm 0.021$ | 6.9/15 | 5/15 |
| ***RGD*** | $\mathbf{0.557 \pm 0.002}$ | $\mathbf{0.545 \pm 0.006}$ | $0.371 \pm 0.019$ | $\mathbf{4.9/15}$ | $\mathbf{4/15}$ |

Table 8: Comparative Results of Proxy Integration with COMs, ROMA, and ours.

| Method | Ant Morphology | TF Bind 10 |
|---|---|---|
| No proxy | $0.940 \pm 0.004$ | $0.657 \pm 0.006$ |
| Vanilla proxy | $0.961 \pm 0.011$ | $0.667 \pm 0.009$ |
| COMs | $0.963 \pm 0.004$ | $0.668 \pm 0.003$ |
| ROMA | $0.953 \pm 0.003$ | $0.667 \pm 0.003$ |
| Ours | $0.968 \pm 0.006$ | $0.694 \pm 0.018$ |

fall short of those achieved by our method as detailed in Table 8. This discrepancy likely stems from the complexity involved in optimizing $y$, whereas dynamically adjusting $\omega$ proves to be a more efficient strategy for enhancing sampling control.

Last but not least, we explore simple annealing approaches for $\omega$. Specifically, we test two annealing scenarios considering the default $\omega$ as 2.0: (1) a decrease from 4.0 to 0.0, (2) an increase from 0.0 to 4.0, both modulated by a cosine function over the time step ($t$) and (3) a high constant $\omega = 4$. We apply these strategies to the Ant Morphology and TF Bind 10 tasks, and the results are as follows:

Table 9: Results of Annealing Approaches.

| Method | Ant Morphology | TF Bind 10 |
|---|---|---|
| RGD | 0.968 | 0.694 |
| $\omega = 2.0$ | 0.940 | 0.657 |
| $\omega = 4.0$ | 0.927 | 0.655 |
| Increase | 0.948 | 0.654 |
| Decrease | 0.924 | 0.647 |

The empirical results across both strategies illustrate their inferior performance compared to our approach, thereby demonstrating the efficacy of our proposed method.

## G   Hyperparameter Sensitivity Analysis

RGD's performance is assessed under different settings of $T$, $y$, and $\eta$. We experiment with $T$ values of 500, 750, 1000, 1250, and 1500, with the default being $T = 1000$. For the condition ratio $y/y_{max}$, we test values of

0.5, 1.0, 1.5, 2.0, and 2.5, considering 1.0 as the default. Similarly, for the learning rate $\eta$, we explore values of $2.5e^{-3}$, $5.0e^{-3}$, 0.01, 0.02, and 0.04, with the default set to $\eta = 0.01$. Results are normalized by comparing them with the performance obtained at default values.

As depicted in Figures 5, 6, and 7, $RGD$ demonstrates considerable resilience to hyperparameter variations. The Ant task, in particular, exhibits a more marked sensitivity, with a gradual enhancement in performance as these hyperparameters are varied. The underlying reasons for this trend include: (1) An increase in the number of diffusion steps ($T$) enhances the overall quality of the generated samples. This improvement, in conjunction with more effective guidance from the trained proxy, leads to better results. (2) Elevating the condition ($y$) enables the diffusion model to extend its reach beyond the existing dataset, paving the way for superior design solutions. However, selecting an optimal $y$ can be challenging and may, as observed in the TFB10 task, sometimes lead to suboptimal results. (3) A higher learning rate ($\eta$) integrates an enhanced guidance signal from the trained proxy, contributing to improved performances.

In contrast, the discrete nature of the TFB10 task seems to endow it with a certain robustness to variations in these hyperparameters, highlighting a distinct behavioral pattern in response to hyperparameter adjustments.

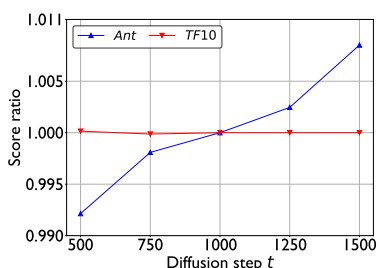

Figure 5: **The ratio of** the performance of our $RGD$ method with $T$ **to** the performance with $T = 1000$.

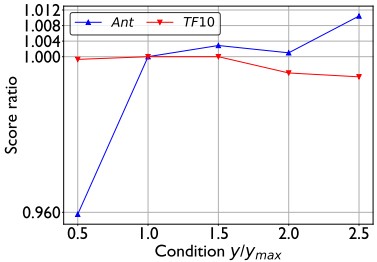

Figure 6: **The ratio of** the performance of our $RGD$ method with $y/y_{max}$ **to** the performance with 1.0.

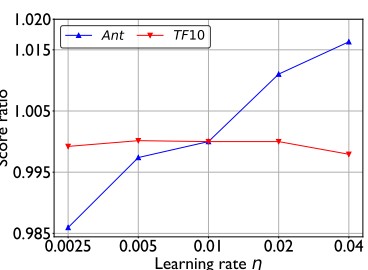

Figure 7: **The ratio of** the performance of our $RGD$ method with $\eta$ **to** the performance with $\eta = 0.01$.

## H    Notations

We provide the key notations used in this paper in Table 10.

## I    Broader Impact and Limitation

**Broader impact.** Our research has the potential to significantly accelerate advancements in fields such as new material development, biomedical innovation, and robotics technology. These advancements could lead to breakthroughs with substantial positive societal impacts. However, we recognize that, like any powerful tool, there are inherent risks associated with the misuse of this technology. One concerning possibility is the exploitation of our optimization techniques to design objects or entities for malicious purposes, including the creation of more efficient weaponry or harmful biological agents. Given these potential risks, it is imperative to enforce strict safeguards and regulatory measures, especially in areas where the misuse of technology could lead to significant ethical and societal harm. The responsible application and governance of such technologies are crucial to ensuring that they serve to benefit society as a whole.

**Limitation.** We recognize that the benchmarks utilized in our study may not fully capture the complexities of more advanced applications, such as protein drug design, primarily due to our current limitations in accessing wet-lab experimental setups. Moving forward, we aim to mitigate this limitation by fostering partnerships with domain experts, which will enable us to apply our method to more challenging and diverse problems. This direction not only promises to validate the efficacy of our approach in more complex scenarios but also aligns with our commitment to pushing the boundaries of what our technology can achieve.

Table 10: Key notations used in this paper.

| Notations | Descriptions |
|---|---|
| $\boldsymbol{\phi}$ | Proxy parameters |
| $\boldsymbol{\theta}$ | Diffusion model parameters |
| $\mathcal{J}_{\boldsymbol{\phi}}(\cdot)$ | Learned mean |
| $\sigma_{\boldsymbol{\phi}}(\cdot)$ | Learned standard deviation |
| $\mathcal{N}$ | Gaussian distribution |
| $p_{\boldsymbol{\phi}}(y\|\hat{\boldsymbol{x}})$ | Proxy distribution |
| $\boldsymbol{x}$ | Particular design |
| $y$ | Property of design |
| $\hat{\boldsymbol{x}}$ | Adversarial designs |
| $q(\boldsymbol{x})$ | Adversarial distribution |
| $t$ | Time step of diffusion sampling |
| $T$ | Total number of diffusion steps |
| $\tau$ | Gradient optimization step on design |
| M | Total number of gradient optimization steps |
| $\boldsymbol{x}_t$ | Noisy sample at time step $t$ |
| $\boldsymbol{f}(\cdot, t)$ | Drift coefficient |
| $g(\cdot)$ | Diffusion coefficient |
| $\boldsymbol{w}$ | Standard Wiener process |
| $\bar{\boldsymbol{w}}$ | Reverse Wiener process |
| $\nabla_{\boldsymbol{x}} \log p(\boldsymbol{x})$ | Score of the marginal distribution |
| $\boldsymbol{s}_{\boldsymbol{\theta}}(\cdot)$ | Learned score function of $\boldsymbol{x}_t$ |
| $x_{d1}/x_{d2}$ | Two-dim variable of Rosenbrock |
| $\mathcal{X}$ | Design space |
| $d$ | Design dimension |
| $\hat{\omega}$ | Optimized strength parameter |
| $\omega$ | Strength parameter |
| $\mathcal{D}$ | Offline dataset |
| $\sigma(t)^2$ | Variance of perturbation kernel |
| $\mu(t)$ | Mean of perturbation kernel |
| $\alpha$ | Hyperparameter of KL loss |

