# OpenReview forum: "Robust Guided Diffusion for Offline Black-Box Optimization"
_TMLR — Accepted by TMLR_

### Review · Reviewer_i1gZ · 2024-10-25

**Summary Of Contributions:**

Offline black-box optimization (BBO) is concerned with the problem of learning the optimum/optimizer of an objective function $J: \\mathbf{x} \\to y$ without explicit access to the function $J$ but rather only implicit knowledge of the objective through a fixed sample of input-output pairs. This limitation of one's knowledge of the objective to a finite sample of data poses a significant challenge to the optimization task.

Most approaches to this problem adopt either a *forward* approach (essentially learning some representation of $p(y| \\mathbf{x})$, such as through regression) or an *inverse* approach (essentially learning $p(\\mathbf{x} | y)$, which is typically a one-to-many mapping). Each approach has its advantages and limitations. This paper proposes Robust Guided Diffusion (RGD), a method for offline BBO that combines elements of both the forward and inverse approaches to leverage their advantages while mitigating their drawbacks. A large focus of the proposed method is on the use of *guidance* in its inverse component by optimizing the guidance weight during the diffusion sampling process. The paper's experimental results show performance matching the state of the art on a number of BBO benchmark tasks.

**Audience:**

Yes

**Broader Impact Concerns:**

I do not have any ethical concerns about this work.

**Claims And Evidence:**

Yes

**Requested Changes:**

* The paper would greatly benefit from an additional editorial pass. In addition to the expository clarity issues mentioned above, there are several passages in which words or punctuation are omitted, and Figure 2 seems to have an incompletely written caption.
* In Algorithm 1, line 4 would read better as "Identify adversarial examples $\\hat{\\mathbf{x}}$ via Eq. 3."
* Two papers [1, 2] come to mind that explore different strategies for adjusting guidance during diffusion sampling, which I believe could be relevant related work, despite their focus on image models. The authors should consider the relevance of this work but should not feel obligated to include it.
* I appreciate the authors' inclusion of the computational overhead involved. I am also sympathetic to the authors' statement that "the time differences between methods for deriving high-performance designs are less critical in actual production environments." Nevertheless, it would be interesting to see how RGD's roughly one-hour requirement for these experiments compares with the time requirements of the competing methods.

[1] Sadat, Seyedmorteza, et al. "CADS: Unleashing the diversity of diffusion models through condition-annealed sampling." arXiv preprint arXiv:2310.17347 (2023).

[2] Kynkäänniemi, Tuomas, et al. "Applying guidance in a limited interval improves sample and distribution quality in diffusion models." arXiv preprint arXiv:2404.07724 (2024).

**Strengths And Weaknesses:**

### **Strengths**

The offline BBO problem is an interesting and challenging one, and the authors do a good job of motivating its importance and applications. The core idea at work, which is to get the "best of both worlds" by using a hybrid forward/inverse approach and tuning the diffusion guidance procedure, makes sense. The experimental results are well reported and support the authors' claim to have developed a method that is robust across a variety of BBO tasks. The ablation studies carried out in the paper are sensible and reasonably comprehensive. The authors have also made the code for the method publicly available.

### **Weaknesses**

While I consider the related work most similar to the current submission quite easy to understand, I had difficulty following some parts of the technical exposition. Some of this was due to my having to tax my working memory while keeping track of the authors' notational choices and terminology for things much more familiar to me under other names. This minor issue went away by the time I gave the paper a second reading, but I was left with the sense that the paper would greatly benefit from an additional editorial pass, both for clarity (preferably by a naïve reader) and general proofreading. When carrying this out, the authors should consider whether the paper is as self-contained as it could be and whether a reasonably skilled reader could implement the described method based on the paper alone.

Here are a couple of examples of where I got hung up:
* In Algorithm 1, line 5 refers to the diffusion distribution in Eq. 12, but it is not clear to me how $p_{\\theta}(\\hat{\\mathbf{x}}|y)$ and $p\_{\\theta}(\\hat{\\mathbf{x}})$ are computed "via the probability flow ODE," since this ODE (which is not actually defined in the paper, while the SDE is) describes the dynamics on $\\mathbf{x}$ and not the probability distributions. That would be a job for the Fokker-Planck equation, but I can't imagine that this is actually what's being computed under the hood.
* On page 5, another proxy is introduced, somewhat confusingly, as $J\_{\\phi}(\\mathbf{x}\_t, t)$, which the authors say is trained to "predict the property of noise $\\mathbf{x}\_t$ at time step $t$." This new proxy shares notation and, apparently, parameterization with the regression model $J_{\\phi}(\\mathbf{x})$, but it is not clear from the surrounding text how this training happens. A direct relationship is drawn between this new proxy and the original (regression) proxy when the authors "reverse the diffusion from $\\mathbf{x}\_t$ back to
$\\mathbf{x}\_0$" in a single step in Eq. 10. But despite the authors' reference, I cannot reconcile their Equation 10 (which bears some resemblance to Tweedie's formula) with anything in Song et al. (2021), nor can I connect it to any well-known relationship between $\\mathbf{x}\_t$ and $\\mathbf{x}\_0$ in diffusion.

These are two examples that are representative of various areas where the technical exposition is insufficiently clear. Of course, if I have misread or missed something in the above cases, I would appreciate it if the authors could set me straight.

### **Bottom Line**

The core ideas in this paper seem sound, and I believe this work would be of interest to the TMLR readership. The experimental results also show that the authors have developed a method that seems to work well on benchmark cases. That the authors have released the code is also a major selling point, which I believe further helps toward fulfilling the claims-and-evidence criterion. However, the paper's clarity issues have me on the fence as to whether this criterion has fully been satisfied. For now, I will give the authors the benefit of the doubt and put down a *qualified* yes on this criterion.

---

> ### Author Response · Authors · 2024-10-26
> **Thanks for your suggestions!**
>
> Dear Reviewer,
>
> Thank you for your valuable feedback. Your insights have been instrumental in enhancing the quality of our paper. We have carefully revised our manuscript based on your suggestions. Changes made in the revised draft are highlighted in red for your convenience.
>
> ### WEAKNESSES
>
> > In Algorithm 1, line 5 refers to the diffusion distribution in Eq. 12, but it is not clear to me how and are computed "via the probability flow ODE," since this ODE (which is not actually defined in the paper, while the SDE is) describes the dynamics on and not the probability distributions. That would be a job for the Fokker-Planck equation, but I can't imagine that this is actually what's being computed under the hood.
>
> Thank you for your insightful comments. In response to your query regarding Algorithm 1, line 5, and the computation of probabilities via the probability flow ODE: We utilize the approach outlined in reference [r1], which derives a neural ODE equivalent to the original SDE. This enables not only sampling from the same distribution as the SDE but also exact likelihood computation. As you mention, this process does incorporate the Fokker-Planck equation, as detailed in Appendix D titled 'Probability Flow ODE' of the reference [r1].
>
> The specific calculations can be found in the function hosted at our repository: https://anonymous.4open.science/r/RGD-27A5/likelihood.py. To clarify this in our manuscript, we propose revising the text to: 'For an adversarial sample $\hat{\boldsymbol{x}} \sim q(\boldsymbol{x})$, we compute $p_{\boldsymbol{\theta}}(\hat{\boldsymbol{x}})$ and $p_{\boldsymbol{\theta}}(\hat{\boldsymbol{x}} | y)$ using the probability flow ODE associated with the SDE, as detailed in Appendix D of [r1]. The implementation can be accessed https://anonymous.4open.science/r/RGD-27A5/likelihood.py. We estimate $p(y)$ using Gaussian kernel-density estimation.' This revision aims to eliminate any ambiguity regarding the computational methods used.
>
> > I cannot reconcile their Equation 10 (which bears some resemblance to Tweedie's formula) with anything in Song et al. (2021), nor can I connect it to any well-known relationship between and in diffusion.
>
> We train the proxy $J_{\phi}(x_0, 0)$ using a straightforward supervised loss as specified in Eq. (2), where $0$ denotes the clean data state. The relationship between $x_0$ and $x_t$ allows for the reconstruction of $x_0$ directly from $x_t$ using the diffusion-derived function $x_0(x_t)$. Consequently, $J_{\phi}(x_t, t) = J_{\phi}(x_0(x_t), 0)$.
>
> We have added a note in the manuscript: 'For a more detailed derivation, please refer to Appendix B.' The Appendix provides a comprehensive derivation of Eq.(10). The derivation is not displayed here due to markdown's formatting limitations.
>
> ### Requested Changes:
>
> > The paper would greatly benefit from an additional editorial pass.
>
> Regarding Figure 2, the caption 'Overall of RGD' is intended to be complete. We are currently reviewing the manuscript to correct any omissions in words and punctuation to ensure the text meets editorial standards.
>
> > In Algorithm 1, line 4 would read better as "Identify adversarial examples via Eq. 3."
>
> We have updated Algorithm 1, line 4, as you suggest.
>
> > Two papers [1, 2] come to mind.
>
> We have reviewed the suggested papers and found their methodologies complementary to our work. In response, we have included a discussion in the revised manuscript:
>
> \citet{sadat2023cads} proposes to add scheduled, monotonically decreasing Gaussian noise to the conditioning vector, to balance sample diversity and condition alignment. \citet{kynkaanniemi2024applying} restricts the guidance to a specific range of noise levels, which improves both sampling speed and quality.
>
> > interesting to see how RGD's roughly one-hour requirement for these experiments compares with the time requirements of the competing methods.
>
> Thank you for your valuable feedback, which emphasizes the importance of computational efficiency in our study. We appreciate the suggestion to compare the time requirements of our RGD method with other competitive methods. For the SuperC task, our method (RGD) requires approximately $3581.3$ seconds. In comparison, Auto.CbAS requires $425.2$ seconds, MIN requires $921.4$ seconds, BONET requires $673.2$ seconds, DDOM requires $460.7$ seconds, and BDI requires $618.4$ seconds. We have incorporated this data into the revised manuscript to provide a clear benchmark of our method's performance.
>
> References:
> - [r1] Song et al. Score-Based Generative Modeling through Stochastic Differential Equations. 2021.
> - [r2] Huang C W, Lim J H, Courville A C. A variational perspective on diffusion-based generative models and score matching. Advances in Neural Information Processing Systems, 2021, 34: 22863-22876.
> - [r3] Ho J, Jain A, Abbeel P. Denoising diffusion probabilistic models[J]. Advances in neural information processing systems, 2020, 33: 6840-6851.

---

> ### Comment · Reviewer_i1gZ · 2024-10-28
> **Response to Authors**
>
> I thank the authors for their rebuttal. I will save any further comments for the full discussion phase but want to quickly address a few points:
> * I found the authors' pointer to the specific portion of the code that implemented the probability-flow ODE helpful.
> * Regarding Eq. 10, it appears that my original interpretation of it as a version of Tweedie's formula was accurate. In my opinion, this also makes it far easier to understand than the reference to the Song et al. appendix. (The Ho et al. reference, now mentioned in the revised appendix, is also easier to grasp than the current reference.) There are a few things that remain in the text that originally confused me and might still confuse other readers. First, Eq. 10 is presented as an *equality*, while the actual relationship is an *approximation*. (The derivation in the revised appendix reflects this, but I recommend making it clearer in the main text.) Eq. 10 actually represents $\\mathbb{E}[\\mathbf{x}\_0 | \\mathbf{x}\_t]$ and not $\\mathbf{x}\_0$. Second, $\\mu(t)$ functions here not as the mean of the perturbation kernel but rather the weight applied to $\\mathbf{x}_0$ at time $t$. The actual mean of the perturbation kernel would then be $\\mu(t) \\mathbf{x}_0$. These are relatively minor points, but they are sufficient to trip up some readers trying to verify the derivations along the way.
> * I am glad that the authors found the suggested related work relevant. In the case of the Sadat et al., paper, the connection to the current work that I had in mind was actually that paper's "Dynamic CFG" (Section 3.3 and Algorithm 2), which adjusts the weight of the CFG term according to a schedule during sampling, although it does not specifically optimize this weight. That the Sadat et al. paper's main condition-annealing method reportedly outperforms that approach is interesting, but it is not as straightforward to connect it to the present paper's proxy-enhanced sampling.
> * With regard to the caption for Figure 2, if this is intended to be complete, I think the authors may be looking for the word *overview* rather than *overall*. The latter word is primarily a modifier and only functions as a noun in common usage when referring to an [article of clothing](https://www.merriam-webster.com/dictionary/overall).
> * I thank the authors for including the computational overhead of competing methods and consider this issue satisfactorily addressed.
> * A few minor typos have been introduced in the new (red) text.
> * As of this writing, the link to the anonymized code in the revised Appendix B directs to an expired repository. The other links that I checked still seem to work. (These links would later be updated to permanent repositories anyway, but I thought I'd mention it in the meantime.)

---

> > ### Author Response · Authors · 2024-10-28
> > **Thank you for your insightful suggestions and thorough review**
> >
> > Thank you for your prompt feedback. We are pleased to know that our responses have effectively addressed your concerns. Based on your suggestions, we have made the necessary revisions to our paper.
> >
> >
> > > Regarding Eq. 10, it appears that my original interpretation of it as a version of Tweedie's formula was accurate. In my opinion, this also makes it far easier to understand than the reference to the Song et al. appendix. (The Ho et al. reference, now mentioned in the revised appendix, is also easier to grasp than the current reference.) There are a few things that remain in the text that originally confused me and might still confuse other readers. First, Eq. 10 is presented as an equality, while the actual relationship is an approximation. (The derivation in the revised appendix reflects this, but I recommend making it clearer in the main text.) Eq. 10 actually represents and not . Second, functions here not as the mean of the perturbation kernel but rather the weight applied to at time. The actual mean of the perturbation kernel would then be. These are relatively minor points, but they are sufficient to trip up some readers trying to verify the derivations along the way.
> >
> > Thank you for your insights. In response to your feedback:
> >
> > - We have incorporated Tweedie's formula into our revised manuscript to provide a clearer context and enhance understanding.
> >
> > - We have modified the presentation of Eq. 10 in our main text from an equality to an approximation, aligning it with the derivation detailed in our revised appendix.
> >
> > - We have updated the description of $\mu(t)$ from "mean of the perturbation kernel" to "mean coefficient of the perturbation kernel," clarifying its role in the perturbation kernel.
> >
> > > I am glad that the authors found the suggested related work relevant. In the case of the Sadat et al., paper, the connection to the current work that I had in mind was actually that paper's "Dynamic CFG" (Section 3.3 and Algorithm 2), which adjusts the weight of the CFG term according to a schedule during sampling, although it does not specifically optimize this weight. That the Sadat et al. paper's main condition-annealing method reportedly outperforms that approach is interesting, but it is not as straightforward to connect it to the present paper's proxy-enhanced sampling.
> >
> > Thank you for your suggestion. We have acknowledged this in our related work section: "\citet{sadat2023cads} introduce the Dynamic CFG, which initially compels the model to depend more on the unconditional score and progressively shifts towards the standard CFG. However, unlike our method, it does not optimize the strength parameter."
> >
> > > With regard to the caption for Figure 2, if this is intended to be complete, I think the authors may be looking for the word overview rather than overall. The latter word is primarily a modifier and only functions as a noun in common usage when referring to an article of clothing.
> >
> >
> > We have updated the caption for Figure 2, replacing "overall" with "overview" to better convey the intended meaning.
> >
> > > A few minor typos have been introduced in the new (red) text.
> >
> > We have corrected several misuses of citet and citep and are continuing to review the document for additional errors to ensure accuracy throughout.
> >
> >
> > > As of this writing, the link to the anonymized code in the revised Appendix B directs to an expired repository. The other links that I checked still seem to work. (These links would later be updated to permanent repositories anyway, but I thought I'd mention it in the meantime.)
> >
> > Thank you for bringing this to our attention. We have updated the link in Appendix B to ensure it directs to the current repository. We appreciate your diligence in checking the links.

---

### Review · Reviewer_rX3i · 2024-10-27

**Summary Of Contributions:**

The paper presents a framework called RGD that combines proxy-based and proxy-free diffusion techniques to enhance black-box optimization tasks. The paper integrates proxy guidance into the diffusion process to improve control and robustness during sample generation, focusing on out-of-distribution (OOD) issues. The authors demonstrate the effectiveness of RGD on a variety of continuous and discrete optimization tasks. The downside of the proposed method is the additional computational overhead required for the refinement process, which appears significant in comparison to other state-of-the-art methods.

**Audience:**

Yes

**Claims And Evidence:**

Yes

**Requested Changes:**

- Please comment more on the computational overhead of the proposed method and compare it on more datasets with respect to related work, with different number and type of features, to provide a better understanding of the trade-off between performance and computational cost also possibly with other hyperparameters like T the number of diffusion steps. I understand your argument that the experiment design's cost is not as significant as the experiment itself, however, the computational overhead currently seems significant and it would be useful to have more data on this to better understand the available trade-offs between algorithmic and hardware performance. The computational cost, since it appears as a significant limitation of the proposed method, should be more prominently mentioned in the main body of the paper.
- It would be useful to strengthen the motivation by pointing out specific scenarios/datasets on which you benchmarked where it is challenging to generate optimal samples, due to the OOD issues, to provide a more concrete benefit of the proposed method
- How did you determine "300 gradient ascent steps" as the optimal number?

Minor:

- It would be useful to have a notation table in the Appendix to be able to quickly reference the symbols used in the paper
- To improve clarity it would be useful to move text from footnotes to the main text where appropriate to avoid breaking the flow of the paper - I would move the details on the datasets/challenges from Experiments to Appendix to make space
- We’ve detailed -> We have detailed

**Strengths And Weaknesses:**

Strengths:
- The paper is clearly written and well-structured, making it easy to follow the proposed framework and experimental results. I especially appreciate Figure 1 for providing a visual motivation of the RGD framework
- Addition of proxy guidance to diffusion-based models to address OOD issues
- The method is compared against state-of-the-art baselines and ablations, providing a comprehensive evaluation of the proposed framework
- The implementation via shared documented code repository enhances reproducibility and transparency

Weaknesses:
- The refinement requires additional computational overhead, which may limit its practicality in resource-constrained environments, notably on the SuperC task it requires 3581.3 seconds in comparison to Auto.CbAS which requires 425.2 seconds, MIN requires 921.4 seconds, BONET requires 673.2 seconds, DDOM requires 460.7 seconds, and BDI requires 618.4 seconds, which is a significant increase in computational cost. This cost should also be more prominently mentioned in the main body of the paper because it is a significant limitation of the proposed method.

---

> ### Author Response · Authors · 2024-10-28
> **Thanks for your suggestions!**
>
> Dear Reviewer,
>
> Thank you for your valuable feedback, which has greatly improved our paper. We have revised the manuscript according to your suggestions, with changes highlighted in red for your convenience.
>
>
> ### Requested Changes:
>
> > Please comment more on the computational overhead of the proposed method.
>
> Our method can complete each task within one hour on a single V100 GPU, a duration we consider manageable. This perspective is especially relevant in offline black-box optimization settings, where the dataset size is generally modest. We do not view this as a significant limitation of our proposed method, as the primary expense lies in the evaluation phase.
>
> In the Appendix E, we discuss the continuous task with a dimension of $86$ (SuperC); here, we further address the discrete $64$-dimension NAS task. Our method (RGD) requires approximately $3191.5$ seconds, while Auto.CbAS takes $389.3$ seconds, MIN takes $879.1$ seconds, BONET takes $485.3$ seconds, DDOM takes $103.4$ seconds, and BDI takes $498.0$ seconds. Evaluating any single method mentioned above involves training the CIFAR-10 dataset over $20$ epochs for $128$ architectural designs, requiring a total of $153.6$ hours. By comparison, the one-hour computation time for our method seems negligible. This contrast is even more stark in the context of biological sequence design, where the evaluation of a single design may span several months.
>
> Regarding hyperparameters such as the number of diffusion steps $T$, their impact on computation time is minimal, as evidenced by the mere $1/100$ ratio shown in Table 4 where $T=1000$. The bulk of the computational effort is attributed to the diffusion-based proxy refinement module, as shown in Table 4. As indicated by the ablation studies in Table 3, even removing this module does not significantly detract from our method’s efficacy.
>
> We have amended the main paper by integrating the following statement at the end of Section 3: "The computational effort of this module can exceed that of the sampling itself, a topic we explore further in Appendix E. Notably, even without this module, our method maintains strong performance, as detailed in Table 3." Additionally, we have included details on the computation costs associated with NAS in Appendix E.
>
> > It would be useful to strengthen the motivation by pointing out specific scenarios/datasets on which you benchmarked where it is challenging to generate optimal samples, due to the OOD issues
>
> Our two modules—proxy-enhanced sampling and diffusion-based proxy refinement—effectively address out-of-distribution (OOD) issues. For the first module, as we discuss in our paper, "Directly updating the design $\boldsymbol{x}_t$ with proxy gradient suffers from the OOD issue. Thus, we propose to introduce proxy guidance by only optimizing the strength parameter $\omega$." The second module "seamlessly integrates insights from proxy-free diffusion back into the proxy for refinement." As detailed in Table 3, using "direct grad update" or removing the diffusion-based proxy refinement module degrades performance across nearly all tasks, demonstrating our method's ability to mitigate OOD issues.
>
> > How did you determine "300 gradient ascent steps" as the optimal number?
>
>
> We do not claim that performing $300$ gradient ascent steps is an optimal choice. As we state in our paper, "We utilize a vanilla proxy to perform $300$ gradient ascent steps, identifying samples with unusually high prediction scores as adversarial."
>
> We conducted $300$ steps to identify a sufficient number of samples with unusually high prediction scores, but we are not asserting this number as optimal. Our primary objective was to detect enough adversarial samples, and the exact number of steps was not critical for this purpose.
>
> > It would be useful to have a notation table in the Appendix to be able to quickly reference the symbols used in the paper
>
> We have incorporated a notation table in Appendix H, titled "Notations," to facilitate quick reference to the symbols used throughout the paper.
>
> > To improve clarity it would be useful to move text from footnotes to the main text where appropriate to avoid breaking the flow of the paper - I would move the details on the datasets/challenges from Experiments to Appendix to make space
>
> We understand the importance of improving the clarity and flow of the paper by appropriately positioning content. However, we believe that many readers may not be familiar with the specific details of our datasets, such as their dimensions and the practical significance of the designs involved. Including this information in the main text is essential for helping readers fully understand our experiments and the context of our work. Therefore, we feel it is necessary to retain these details in the main body of the paper to enhance comprehension and accessibility for a broader audience.
>
> > We’ve detailed -> We have detailed
>
> We have changed this.

---

> > ### Comment · Reviewer_rX3i · 2024-10-28
> > **Thanks for response**
> >
> > I thank the authors for their swift response.
> >
> > I see regarding the computational expense. Then, if the other methods had the same computational budget as yours e.g. 3191.5 seconds for Auto.CbAS instead of 389.3 seconds, roughly $8\times$ more compute time available for the other method, would your method still be the best among the others?

---

> > > ### Author Response · Authors · 2024-10-28
> > > **Thanks for your insightful question**
> > >
> > > Dear Reviewer,
> > >
> > > Thank you for your prompt feedback. Scaling up other methods such as Auto.CbAS is not straightforward. Auto.CbAS employs a dual-model architecture consisting of a predictor with a hidden size of $2048$ and a VAE model with a hidden size of $256$. When we experimented with increasing the hidden dimensions—multiplying the predictor size to $16384$ ($2048$ x $8$) and the VAE size to $2048$ ($256$ x $8$)—we observed a decline in performance: from $0.421$ to $0.415$ on SuperC and from $0.506$ to $0.497$ on NAS. We attribute this degradation to the limited training data, which likely causes the enlarged models to overfit.
> > >
> > > To address your concerns regarding computational costs, we modified our method by removing the diffusion-based proxy refinement module. As detailed in Table 4, this adjustment significantly reduces computational overhead: from $3581.3$ seconds to $476.7$ seconds on SuperC, and from $3191.5$ seconds to $95.3$ seconds on NAS, rendering our method more efficient than the comparison methods. Following this adjustment, we recalculated the rankings based on the results presented in Tables 1, 2, and 3. The new ranking, as shown in the table below, reaffirms that our method continues to hold its position as the top-performing method in terms of ranking. We have included the updated rankings in our paper.
> > >
> > > | Method            | Rank Mean | Rank Median |
> > > |-------------------|-----------|-------------|
> > > | BO-qEI            | $9.14$    | $11.00$     |
> > > | CMA-ES            | $7.14$    | **3.00**      |
> > > | REINFORCE         | $11.29$   | $14.00$     |
> > > | Grad              | $9.00$    | $10.00$     |
> > > | COMs              | $10.00$   | $10.00$     |
> > > | ROMA              | $5.86$    | $6.00$      |
> > > | NEMO              | $5.14$    | $5.00$      |
> > > | IOM               | $7.43$    | $6.00$      |
> > > | BDI               | $9.29$    | $8.00$      |
> > > | CbAS              | $8.57$    | $8.00$      |
> > > | Auto.CbAS         | $10.29$   | $10.00$     |
> > > | MIN               | $10.29$   | $10.00$     |
> > > | BONET             | $7.43$    | $7.00$      |
> > > | DDOM              | $4.71$    | $5.00$      |
> > > | Ours w/o diffusion-b r | **3.71** | **3.00** |

---

> > > > ### Comment · Reviewer_rX3i · 2024-10-28
> > > > **Thank you**
> > > >
> > > > Thank you! Now I see and understand the compute efficiency with respect to the one-time cost. nit: The new Table 5 misses a period at the end of the caption.

---

> > > > > ### Author Response · Authors · 2024-10-28
> > > > > **Thank you.**
> > > > >
> > > > > We are glad to have clarified your concerns. We have also corrected the caption of Table 5.

---

### Review · Reviewer_pK3b · 2024-11-01

**Summary Of Contributions:**

This work proposes a new optimization framework based on a guided diffusion model. The task at hand is optimization of the desired value metric $y$ given measured designs $x$ and the authors utilize a classifier-free diffusion model $s_\theta(x_t, y)$, updating the solution over time steps $t=T-1,\dots,0$ and returning a design $x_0$ with the maximized value. The authors use the term "proxy-free diffusion" throughout to highlight their application to regression as opposed to classification.

The authors propose two novel modules incorporated into this process: (i) *proxy-enhanced sampling*, which improves the quality of generated samples by leveraging the proxy $J_\phi(x)$ that predicts $y$ and (ii) *diffusion-based proxy refinement*, which modifies the proxy distribution to be more robust against adversarial examples.

Specifically, the *proxy-enhanced sampling* module optimizes the parameter $w$ in the classifier-free guidance score, which balances sample diversity with conformity to the designs with the highest $y$. At each timestep $t$, the previous sample $x_{t+1}$ is updated using the classifier-free score with the current weight $w$, then $w$ is updated to $\hat{w}$ using the gradient of the proxy $J_\phi(x_t(w))$, and finally $x_{t+1}$ is updated again using $\hat{w}$. The *diffusion-based proxy refinement* module regularizes the loss fitting the proxy distribution $p_\phi(y|x)$ (parameterized by $J_\phi$) with a KL-divergence term between the proxy distribution and the diffusion distribution evaluated at adversarial samples, i.e. samples for which the proxy predicts unusually high $y$ values. The proxy is refined prior to the diffusion process.

**Audience:**

Yes

**Broader Impact Concerns:**

No concerns

**Claims And Evidence:**

Yes

**Requested Changes:**

Critical requested changes:
- Frame *Introduction*, *Preliminaries* and *Method* within the forward/inverse approach dichotomy (does the proposed framework follow the inverse approach?). Define how guided diffusion relates to diffusion models and how proxy diffusion and proxy-free diffusion (as well as proxy distribution and diffusion distribution) relate to the forward/inverse approach.
- Modify Figure 2 to have a clear flow of the inference corresponding to Algorithm 1, denote the proxy distribution and proxy-free diffusion model in Figure 2.
- Unify the *Method* section, Figure 2, and Algorithm 1 to present the steps of RGD in the same order to improve clarity.
- Define probability flow ODE and explain whether the proposed framework involves both forward and reverse diffusion.
- Restructure *Related Work* to appear prior to or combine with Section 4.2.
- Modify the nearly identical sentences describing the novel modules to convey more information about the method (e.g. a sentence "proxy-enhanced sampling integrates proxy guidance to enable enhanced sampling control" appears in a nearly identical way 5 times in the manuscript).
- Give more details on adversarial samples and how they manifest as a weakness of the proxy distribution. Provide more context for the inability of the proxy-free diffusion to generate samples beyond the training distribution and why this setting is particularly applicable to offline BBO.
- Explain why $J_\phi(x_t, t)$, which is used for optimizing $w$, is defined w.r.t. $x_0$.
- Provide more details on the "direct grad update" setting in the ablation studies.
- Report brief results of the supplementary analyses given in Appendices E, F in the main text and specify why those two tasks were selected.

Minor requested changes:
- Specify what abbreviations stand for (MINs, GAN, COMs).
- Figure 2 caption: Overall -> Overview
- Provide citations when introducing diffusion models.
- Section 3.2, *Proxy Refinement*: $\mathcal{D}$ notation for the KL divergence is overloaded as it has previously denoted the dataset.
- Section 3.2, *Proxy Refinement*: parameterization trick -> reparameterization trick
- Provide more details on the bi-level optimization of $\alpha$ hyperparameter in terms of the order of operations and the use of training vs. validation set.
- Section 4.1: footnote 4 seems to be in the wrong place.
- Section 4.2: *Comparison Methods* -> *Method Comparison*
- Expand the statement "we have tailored our training methodologies for all approaches" and include more details on the training procedure in the Appendix.
- Add more detailed captions for the figures, especially Figures 3, 4.
- Specify in the caption what is reported (e.g. mean +- std) and bolded for all Tables.

Questions for authors:
- How much value does explicit optimization of $w$ provide over simply setting $w$ to a constant high value? Does one of the benchmarked methods implement this approach and thus could provide an empirical answer to this?
- How is the unconditional score $s_\theta(x_t)$ computed from the neural network  $s_\theta(x_t, y)$?
- Section 3.1, *Proxy Training*: given the for-loop in the Algorithm 1, should it be initial time step $t=T$ instead of $t=0$?

**Strengths And Weaknesses:**

Strengths
- The authors outline all steps of the proposed framework, present unified notation, and provide equations and derivations underlying all steps.
- The authors perform comprehensive benchmarking of their method on a range of continuous and discrete tasks and against a suite of existing methods that follow traditional, forward, and inverse approaches.
- The proposed framework includes novel modules with sound theoretical formulation that are empirically demonstrated to perform well on a range of benchmarks and in ablation studies.

Weaknesses
- The order of presentation in the manuscript does not facilitate clarity. Figures and equations are referenced before they have appeared in the main text; *Related Work* appears at the end and disconnected from the *Preliminaries* and *Comparison Methods* sections; concepts are introduced without clearly relating to each other (e.g. forward/inverse approach, proxy diffusion and proxy-free diffusion, diffusion model, guided diffusion, and diffusion distribution).
- The issues that the two proposed modules aim to address are described in similar terms: the proxy is susceptible to the out-of-distribution issue and overestimation, while proxy-free diffusion struggles with generating samples exceeding the training distribution. Describing the differences between these two settings more clearly would help with motivating the method.
- The authors included additional analyses in the Appendix and briefly mention their existence in Sections 4.5 and 4.6. However, the results of these analyses are not reported in the main text. While details and figures of supplementary analyses may be included in the appendix, all results must be reported in the main text.

---

> ### Author Response · Authors · 2024-11-01
> **Thanks for your suggestions!**
>
> Dear Reviewer,
>
> We sincerely appreciate your valuable feedback, which has significantly enhanced our paper. We have revised the manuscript in line with your suggestions, with all changes highlighted in red for your convenience.
>
> ### Requested Changes
>
> > Frame Introduction, Preliminaries and Method within the forward/reverse approach dichotomy (does the proposed framework follow the reverse approach?). Define how guided diffusion relates to diffusion models and how proxy diffusion and proxy-free diffusion (as well as proxy distribution and diffusion distribution) relate to the forward/reverse approach.
>
> 1. To clarify, the proposed framework adheres to the reverse approach.
>
> 2. While diffusion models capture the natural data distribution, guided diffusion incorporates a guidance mechanism to produce samples with specific desired properties. In proxy diffusion, this guidance mechanism is achieved through a trained proxy. Conversely, in proxy-free diffusion, guidance is implemented via an additional conditional diffusion model.
>
> 3. As highlighted in the third paragraph of the introduction, proxy-free diffusion aligns with the reverse approach. In contrast, proxy diffusion is more consistent with the forward approach, as it uses proxy gradients to iteratively refine samples.
>
> 4. To further clarify, we can conceptualize the mean of the proxy distribution as the proxy, with the forward approach relying on explicit gradients from the proxy to optimize existing designs. The diffusion distribution, in contrast, is derived from our proxy-free diffusion, representing an reverse approach.
>
> The manuscript has been updated to incorporate these clarifications.
>
> > Modify Figure 2 to have a clear flow of the inference corresponding to Algorithm 1, denote the proxy distribution and proxy-free diffusion model in Figure 2.
>
> In Figure 2, we illustrate the proxy $J(x)$, which represents the mean of the proxy distribution. Since only the proxy is used within the proxy-enhanced sampling module, we display only the proxy here to maintain clarity.
>
> As discussed, our method RGD operates as a proxy-free diffusion framework; thus, both (a) and (b) in Figure 2 represent the proxy-free diffusion model.
>
> > Unify the Method section, Figure 2, and Algorithm 1 to present the steps of RGD in the same order to improve clarity.
>
>
> We recognize that the current order may cause confusion: we first present diffusion-based proxy refinement in Algorithm 1, followed by proxy-enhanced sampling, while the Method section presents these in reverse. For Algorithm 1, we must retain the actual execution order, so this sequence cannot be modified. In the Method section, our narrative is as follows: since proxy-free diffusion lacks explicit guidance, we introduce a proxy to enhance the sampling process of the proxy-free diffusion model. After establishing the proxy, we then describe how to improve it using diffusion-based proxy refinement. Presenting diffusion-based proxy refinement first is not feasible, as the proxy is introduced only with the proxy-enhanced sampling module.
>
> > Define probability flow ODE and explain whether the proposed framework involves both forward and reverse diffusion.
>
>
> In our context, the diffusion model is defined by an SDE, and prior work~[r1] derives an equivalent neural probability flow ODE that samples from the same distribution as the SDE. We utilize the probability flow ODE here because it allows for exact likelihood computation for specific samples, as discussed in Section 3.2, Diffusion-based Proxy Refinement.
>
> The proposed framework comprises two modules: proxy-enhanced sampling and diffusion-based proxy refinement. The first module involves enhancing the reverse diffusion sampling process via explicit proxy guidance. The second module leverages the probability flow ODE to compute the diffusion distribution, which can be considered as involving the forward diffusion process.
>
>        [r1] Song Y, Sohl-Dickstein J, Kingma D P, et al. Score-based generative modeling through stochastic differential equations[J]. arXivpreprint arXiv:2011.13456, 2020.
>
> > Restructure Related Work to appear prior to or combine with Section 4.2.
>
>
> We have relocated Related Work to appear after Preliminaries and before Method.
>
> > Modify the nearly identical sentences
>
>
> The repetition was intentional to help readers, especially those whose native language is not English, become familiar with the term.
>
> To address the concern, we have varied the phrasing, for example, changing "enable enhanced sampling control" to "facilitate improved sampling control" to add diversity.

---

> > ### Author Response · Authors · 2024-11-01
> >
> > > Give more details on adversarial samples and how they manifest as a weakness of the proxy distribution. Provide more context for the inability of the proxy-free diffusion to generate samples beyond the training distribution and why this setting is particularly applicable to offline BBO.
> >
> >
> > In the Method section, we state: "We utilize a vanilla proxy to perform $300$ gradient ascent steps, identifying samples with unusually high prediction scores as adversarial. This method is based on the limited extrapolation capability of the vanilla proxy, as demonstrated in Figure 3 in COMs[r2]. ".
> >
> > These adversarial samples are out-of-distribution with respect to the proxy distribution.
> >
> > Besides Figure 1 to illustrate the limitations of proxy-free diffusion, we have already included the results from the w/o proxy-e ablation in Table 3 further highlight its limitations.
> >
> > This setting is particularly relevant to offline BBO because the offline dataset is inherently limited, and we aim to find samples with desired properties that extrapolate beyond the dataset.
> >
> >     [r2] Trabucco B, Kumar A, Geng X, et al. Conservative objective models for effective offline model-based optimization[C]//International Conference on Machine Learning. PMLR, 2021: 10358-10368.
> >
> > > Explain why, which is used for optimizing, is defined w.r.t..
> >
> > Ideally, $J(x_t, t)$ should be defined with respect to $x_t$, but this would require proxy training across different time steps, which is computationally intensive. To reduce computational cost, we train only $J(x_0)$ and then use the relationship between $x_0$ and $x_t$ to derive $J(x_t, t)$, as detailed in Eq. (10).
> >
> > > Provide more details on the "direct grad update" setting in the ablation studies.
> >
> > We perform gradient ascent directly on the diffusion intermediate state $x_t$ using the gradient of $J(x_t, t)$, with a learning rate of $0.1$.
> >
> > > Report brief results of the supplementary analyses given in Appendices E, F in the main text and specify why those two tasks were selected.
> >
> >
> > We have revised the draft to include brief results from the supplementary analyses in Appendices E and F.
> >
> > Ant and TF10 were chosen because they both offer ground-truth oracle evaluation rather than relying on a fitted neural network as the oracle. Ant represents a typical continuous task, while TF10 serves as a standard example of a discrete task.
> >
> > ### Minor requested changes:
> >
> > > Specify what abbreviations stand for (MINs, GAN, COMs).
> >
> > The abbreviations are derived from the respective papers. When we mention these abbreviations, we have already cited their corresponding sources. Specifically, MINs stands for Model Inversion Networks, GAN stands for Generative Adversarial Networks, and COMs stands for Conservative Objective Models.
> >
> > > Figure 2 caption: Overall -> Overview
> >
> > We have made this change.
> >
> > > Provide citations when introducing diffusion models.
> >
> > We have added citations for diffusion models in the Preliminaries section.
> >
> > >  notation for the KL divergence is overloaded as it has previously denoted the dataset.
> >
> > We have changed the KL divergence notation to $\text{KL}$ instead of using $D$.
> >
> > > Section 3.2, Proxy Refinement: parameterization trick -> reparameterization trick
> >
> > We have made this change.
> >
> > > Provide more details on the bi-level optimization of
> >  hyperparameter in terms of the order of operations and the use of training vs. validation set.
> >
> > The inner optimization task is efficiently approximated via first-order gradient descent methods. We use batch optimization, with each batch containing $256$ training samples and $256$ validation samples. The bi-level optimization process updates the hyperparameter with a single iteration for both the inner and outer levels. We have revised our paper to incorporate these changes.
> >
> > > Section 4.1: footnote 4 seems to be in the wrong place.
> >
> > We have removed this footnote as the task no longer presents any issues.
> >
> > > Section 4.2: Comparison Methods -> Method Comparison
> >
> > We have made this change.
> >
> > > Expand the statement "we have tailored our training methodologies for all approaches"
> >
> > We have tailored our training methodologies for all approaches, utilizing a three-layer MLP architecture for all involved proxies.
> >
> > > Add more detailed captions for the figures, especially Figures 3, 4.
> >
> > We have added more detailed captions for Figures 3 and 4.
> >
> > > Specify in the caption what is reported (e.g. mean +- std) and bolded for all Tables.
> >
> > We have already clarified the metrics in the Experimental Configuration section. We attempted to add (mean ± std) and other details in the tables, but doing so made the tables visually cluttered.

---

> > > ### Author Response · Authors · 2024-11-01
> > >
> > > ### Questions for authors:
> > >
> > >
> > > > How much value does explicit optimization of
> > >  provide over simply setting
> > >  to a constant high value? Does one of the benchmarked methods implement this approach and thus could provide an empirical answer to this?
> > >
> > > In our original submission, we evaluated (1) a decrease from $4.0$ to $0.0$ and (2) an increase from $0.0$ to $4.0$, both modulated by a cosine function over the time step ($t$). Experiments were conducted on two tasks, Ant and TF10, with results reported in Table 9.
> > >
> > > We have now additionally tested a constant $w = 4.0$ and included these results in Table 9. The constant $w = 4.0$ reduces performance, demonstrating the effectiveness of our proposed approach.
> > >
> > > > How is the unconditional score
> > >  computed from the neural network ?
> > >
> > > We compute the unconditional score $s(x_t)$ by setting the label $y = 0$, calculating it as $s(x_t, y = 0)$. This works because we normalize the labels to a standard Gaussian distribution with a zero mean, so a zero label indicates the unconditional case.
> > >
> > > > Section 3.1, Proxy Training: given the for-loop in the Algorithm 1, should it be initial time step t=T instead t=0?
> > >
> > > In the diffusion setting, $t = 0$ represents clean data, and $t = T$ represents pure noise.
> > >
> > > In the sampling process of Algorithm 1, we start from pure noise at $t = T$ and transform it into clean data at $t = 0$ with desired properties.
> > >
> > > For proxy training, we train the proxy $J(x_0)$ on clean data $x_0$ and derive $J(x_t)$ from $J(x_0)$.

---

> > > > ### Comment · Reviewer_pK3b · 2024-11-07
> > > > **Response to Official Comment by Authors**
> > > >
> > > > I thank the authors for their clarifications and the changes made to the manuscript.
> > > >
> > > > I propose the following minor changes which in my view would further enhance the clarity of the manuscript:
> > > >
> > > > - Section 2.1: "A common approach gradient ascent..." -> "A forward approach for BBO uses gradient ascent to fit a proxy distribution...". This change would explicitly connect this section to the second paragraph of the Introduction.
> > > >
> > > > - Section 2.3: "Proxy-free guidance, not dependent on proxy gradients, enjoys an inherent robustness..." -> "In proxy-free diffusion, guidance is not dependent on proxy gradients, which enables an inherent robustness..."
> > > >
> > > > - Section 4.1: "This is particularly significant in offline BBO where..." -> "This is particularly significant in offline BBO where we aim to obtain samples beyond the training distribution as the offline dataset is inherently limited."
> > > >
> > > > - Section 4.1, Proxy Training: "This distribution is trained exclusively at the initial time step..." -> "This distribution is trained exclusively at the initial time step t = 0, eliminating the need for training across time steps and reducing computational cost."
> > > >
> > > > ### Remaining requests
> > > >
> > > > - Figure 2: I appreciate the authors' clarification on the order of their narrative and propose to include it in the Figure 2 caption similar to the following (leaving the precise wording to the authors):
> > > > "Overview of RGD. Since proxy-free diffusion lacks explicit guidance, we use a proxy to enhance the sampling process of our proxy-free diffusion model (a). We ensure that the proxy is safeguarded against adversarial samples by using the diffusion distribution to refine it (b)."
> > > >
> > > > - Introduction: "However, this technique is susceptible to the out-of-distribution (OOD) issue, leading to potential overestimation of unseen designs and resulting in adversarial solutions (Trabucco et al., 2021)." Please briefly define here what an adversarial solution is. The definition of adversarial samples provided in the Methods is too far into the main text.
> > > >
> > > > - Please add the following sentence to the caption of Tables 1,2,3,6,7: "Values are mean ± SE." Adding a short sentence to the caption will help a reader see what is reported at a glance and should not add visual clutter.
> > > >
> > > > - Please also briefly report in the main text results of the hyperparameter sensitivity analysis detailed in Appendix G.

---

> > > > > ### Comment · Reviewer_pK3b · 2024-11-07
> > > > > **Remaining questions for the authors**
> > > > >
> > > > > Could you please clarify the difference between the "w/o proxy-e" and "direct grad update" ablation scenarios in terms of which steps in Algorithm 1 are omitted in each scenario?
> > > > >
> > > > > > We have tailored our training methodologies for all approaches, utilizing a three-layer MLP architecture for all involved proxies.
> > > > >
> > > > > Are the training methodologies for all approaches compared in this work reported in sufficient detail for a reader to reproduce the obtained results? Is the code implementing methods other than RGD released?
> > > > >
> > > > > > This works because we normalize the labels to a standard Gaussian distribution with a zero mean, so a zero label indicates the unconditional case.
> > > > >
> > > > > Is this modeling choice of normalizing labels to a standard normal a commonly performed strategy for the considered design-bench tasks or would it be beneficial to include this in *Methods* for reproducibility?

---

> > > > > > ### Author Response · Authors · 2024-11-07
> > > > > >
> > > > > > > Could you please clarify the difference between the "w/o proxy-e" and "direct grad update" ablation scenarios in terms of which steps in Algorithm 1 are omitted in each scenario?
> > > > > >
> > > > > > To clarify, the 'w/o proxy-e' ablation scenario omits steps 12 and 13 in Algorithm 1. Similarly, the 'direct grad update' scenario also removes steps 12 and 13 but differs by updating xt directly using the proxy gradient.
> > > > > >
> > > > > > > Are the training methodologies for all approaches compared in this work reported in sufficient detail for a reader to reproduce the obtained results? Is the code implementing methods other than RGD released?
> > > > > >
> > > > > > Yes, the comparison methods we use have open-source implementations. Most baselines can be found at: https://github.com/brandontrabucco/design-baselines.
> > > > > >
> > > > > > > Is this modeling choice of normalizing labels to a standard normal a commonly performed strategy for the considered design-bench tasks or would it be beneficial to include this in Methods for reproducibility?
> > > > > >
> > > > > > Yes, normalizing labels to a standard normal is a common strategy, as used in the seminal work, design-bench [r1].
> > > > > >
> > > > > >      [r1] Trabucco B, Geng X, Kumar A, et al. Design-bench: Benchmarks for data-driven offline model-based optimization[C]//International Conference on Machine Learning. PMLR, 2022: 21658-21676.

---

> > > > > ### Author Response · Authors · 2024-11-07
> > > > >
> > > > > Thank you for your valuable suggestions. We have implemented most of them in our revisions.
> > > > >
> > > > > > Figure 2: I appreciate the authors' clarification on the order of their narrative and propose to include it in the Figure 2 caption similar to the following (leaving the precise wording to the authors): "Overview of RGD. Since proxy-free diffusion lacks explicit guidance, we use a proxy to enhance the sampling process of our proxy-free diffusion model (a). We ensure that the proxy is safeguarded against adversarial samples by using the diffusion distribution to refine it (b)."
> > > > >
> > > > > We have revised the caption to provide clarity as follows: "Overview of RGD: Module (a) incorporates proxy guidance into proxy-free diffusion to enable enhanced sampling control; Module (b) integrates insights from proxy-free diffusion back into the proxy for refinement."
> > > > >
> > > > > > Introduction: "However, this technique is susceptible to the out-of-distribution (OOD) issue, leading to potential overestimation of unseen designs and resulting in adversarial solutions (Trabucco et al., 2021)." Please briefly define here what an adversarial solution is. The definition of adversarial samples provided in the Methods is too far into the main text.
> > > > >
> > > > > We understand the importance of clarity in the introduction. However, we believe the concept of adversarial solutions is widely recognized within both the offline BBO and broader ML communities. Additionally, the phrase 'overestimation of unseen designs' in the sentence already hints at the nature of adversarial solutions, and we have provided a relevant citation for further context. Including an additional definition here may result in redundant wording. We hope this addresses your concern while maintaining the conciseness of the introduction.
> > > > >
> > > > > > Please add the following sentence to the caption of Tables 1,2,3,6,7: "Values are mean ± SE." Adding a short sentence to the caption will help a reader see what is reported at a glance and should not add visual clutter.
> > > > >
> > > > > We have updated the captions for Tables 1, 2, 3, 6, and 7 to include 'mean ± std.' This addition provides clarity at a glance while maintaining a clean presentation.
> > > > >
> > > > > > Please also briefly report in the main text results of the hyperparameter sensitivity analysis detailed in Appendix G.
> > > > >
> > > > > We have updated the main text to include a brief summary of the hyperparameter sensitivity analysis.

---

> > > > > > ### Comment · Reviewer_pK3b · 2024-11-08
> > > > > > **Response to the official comment**
> > > > > >
> > > > > > Thank you for your clarifications. Did the authors post a new revision? I do not see the referenced changes in the current version.

---

> > > > > > > ### Author Response · Authors · 2024-11-08
> > > > > > >
> > > > > > > We just posted it. Apologies for the oversight earlier.

---

> > > > > > > > ### Comment · Reviewer_pK3b · 2024-11-09
> > > > > > > > **Response to revision**
> > > > > > > >
> > > > > > > > Thank you for the revision.
> > > > > > > >
> > > > > > > > - In Figure 2 caption, there is a typo in "refinement".
> > > > > > > > - In the main text, it is stated that "mean values and standard errors" are reported in the Tables. "std" typically refers to the standard deviation, not standard error, so the addition "mean ± std" to Table captions is not quite correct.
> > > > > > > > - Are the authors planning to incorporate the four minor changes specified [here](https://openreview.net/forum?id=4JcqmEZ5zt&noteId=brwT5IyzQm)? I suggested these additions as the authors' rebuttal made several things clearer to me and it seemed valuable to include those in the main text.

---

> > > > > > > > > ### Author Response · Authors · 2024-11-13
> > > > > > > > >
> > > > > > > > > Thank you for your valuable suggestions!
> > > > > > > > >
> > > > > > > > > (1) We have corrected the typo in "refinement" in the Figure 2 caption.
> > > > > > > > >
> > > > > > > > > (2) We have updated the main text to replace "mean values and standard errors" with "mean values and standard deviations" to ensure accuracy and consistency.
> > > > > > > > >
> > > > > > > > > (3) We have incorporated the four minor changes you recommended, as we agree that these additions provide clarity to the points raised in the rebuttal and enhance the overall quality of the main text.

---

### Decision · Action_Editor_NfdL · 2024-12-02

**Recommendation:** Accept as is

**Comment:**

The paper describes a new sampling scheme for conditional sampling for offline black-box optimization. The claims of the paper are well supported by experiments. Reviewers pointed out some clarifications issues in the first version, which however were resolved during the rebuttal.

**Audience:**

The paper explores the inverse approach to black-box optimization, which heavily relies on machine learning techniques. Therefore, I believe this topic is relevant to certain segments of the TMLR audience.

**Claims And Evidence:**

According to the reviewer, the paper supports all claims with empirical evidence.